# A structural rationale for reversible *vs* irreversible amyloid fibril formation from a single protein

Lukas Frey [1,7], Jiangtao Zhou [2,7] ✉, Gea Cereghetti [3,4], Marco E. Weber [1], David Rhyner[1], Aditya Pokharna[1], Luca Wenchel[1], Harindranath Kadavath [1], Yiping Cao[5], Beat H. Meier[1], Matthias Peter [3], Jason Greenwald[1], Roland Riek [1] ✉ & Raffaele Mezzenga [2,6] ✉

Reversible and irreversible amyloids are two diverging cases of protein (mis) folding associated with the cross-β motif in the protein folding and aggregation energy landscape. Yet, the molecular origins responsible for the formation of reversible vs irreversible amyloids have remained unknown. Here we provide evidence at the atomic level of distinct folding motifs for irreversible and reversible amyloids derived from a single protein sequence: human lysozyme. We compare the 2.8 Å structure of irreversible amyloid fibrils determined by cryo-electron microscopy helical reconstructions with molecular insights gained by solid-state NMR spectroscopy on reversible amyloids. We observe a canonical cross-β-sheet structure in irreversible amyloids, whereas in reversible amyloids, there is a less-ordered coexistence of β-sheet and helical secondary structures that originate from a partially unfolded lysozyme, thus carrying a "memory" of the original folded protein precursor. We also report the structure of hen egg-white lysozyme irreversible amyloids at 3.2 Å resolution, revealing another canonical amyloid fold, and reaffirming that irreversible amyloids undergo a complete conversion of the native protein into the cross-β structure. By combining atomic force microscopy, cryo-electron microscopy and solid-state NMR, we show that a full unfolding of the native protein precursor is a requirement for establishing irreversible amyloid fibrils.

Amyloids were initially identified in the context of severe pathologies such as Parkinson's, Alzheimer's and Huntington's diseases, along with many other neurodegenerative or systemic disorders[1–3]. Later, functional amyloids with diverse biological functionalities were also discovered in many organisms, from bacteria to humans[4–8], and today artificial amyloids are routinely produced from almost any protein to serve a multitude of applications[3,9,10]. The hallmark of all amyloids is their characteristic fibrillar cross-β structure in which β-strands that run perpendicular to the fibril axis stack with a ~4.7 Å spacing in the direction of the fibril axis resulting in a corrugated supramolecular foil composed of β-sheets. These indefinitely long sheets whose length runs parallel to the fibril axis densely pack parallel to each other with a typical spacing of ~10 Å, thereby forming the core of the amyloids. The packing is generally very tight and is stabilized by supramolecular

[1]Institute of Molecular Physical Science, ETH Zürich, Vladimir-Prelog-Weg 2, Zürich, Switzerland. [2]ETH Zurich, Department of Health Sciences and Technology, Zurich, Switzerland. [3]Institute of Biochemistry, Department of Biology, ETH Zurich, Zurich, Switzerland. [4]University of Cambridge, Department of Chemistry, Lensfield Road, Cambridge, United Kingdom. [5]Department of Food Science and Technology, School of Agriculture and Biology, Shanghai Jiao Tong University, Shanghai, China. [6]ETH Zurich, Department of Materials, Zurich, Switzerland. [7]These authors contributed equally: Lukas Frey, Jiangtao Zhou. ✉e-mail: jiangtao.zhou@hest.ethz.ch; roland.riek@phys.chem.ethz.ch; raffaele.mezzenga@hest.ethz.ch

H-bonding interactions between amide proton and carbonyl oxygen atoms in the peptide backbone. Their robustness and stability is further reinforced by hydrophobic contacts between the β-sheets, leading to a packing motif often referred to as steric zipper[11], which is the structural signature of these ultra-stable amyloids. This steric zipper motif is frequently observed in pathological amyloids and endows them with exceptional energetic stability and low free energy of formation, which can be as low as −20 kcal/mol per molecule or −0.4 kcal/mol per residue[12,13], and has largely contributed to the general assumption that they are all irreversible proteinaceous aggregates.

More recently a new class of reversible amyloids has started to emerge encompassing RNA-binding proteins such as FUS, hnRNPA1, hnRNPA2, pyruvate kinase and TDP-43[13–18], which owe their labile nature to constitutive low-complexity domains (LCDs), also referred as low-complexity, amyloid-like, reversible, kinked segments (LARKS)[19,20]. Compared to irreversible amyloids based on the steric zipper motif, reversible amyloid fibrils represent more dynamic states of protein assemblies, and thus tend to have a lower energetic stability and higher free energies, of the order of −10 kcal/mol per molecule or −0.2 kcal/mol per residue[12,13]. Many of these reversible amyloids play functional roles in the organism, such as the stress granules composed of RNA-binding proteins including FUS, hnRNPA2, and pyruvate kinase[6,14] or hormone amyloids such as β-endorphin, and their stability can be modulated to regulate their biological function[21,22].

In recent years, the consensus has grown towards a sequence-specificity paradigm, where the primary sequence of the amyloid-forming peptide or protein is believed to determine the formation of either irreversible-like steric zipper or reversible-like LARKS motifs[13,18]. Accordingly, it was also hypothesized that mutations of the primary sequence might favour reversible amyloids to switch to irreversible ones. This hypothesis applies to most current reversible amyloids ranging from pyruvate kinase[14], hnRNPA1[18] and hnRNPA2[13,18], to TDP-43 and FUS[23]. This view was, however, challenged by a recent report from our group[24] in which human and hen egg-white lysozyme (HEWL), two proteins with 76% sequence identity, were both found to be capable of folding into distinct amyloids, one flexible and reversible upon thermal treatment, and the other rigid and irreversible. These two amyloid types are formed under similar unfolding and fibrillization conditions and without any post-translational modification of the primary sequence. This finding created an apparent conundrum in which a single protein sequence, which is typically viewed as having a single lowest-energy state, may in fact fold into both irreversible and reversible amyloids, each presumably having distinct amyloid motifs, leaving unchanged its primary sequence. However, the indications of different folding motifs for these two extremes has until now only been provided indirectly via their circular dichroism or FTIR spectra[24]. Despite their significantly different mesoscopic polymorphisms, no clear picture exists at the atomic scale and so the molecular mechanisms that govern the reversible vs irreversible selection for amyloid folding from a single protein remain to be fully elucidated.

Here we go beyond the mesoscopic polymorphisms of amyloids from human lysozyme and HEWL, providing unprecedented evidence at the atomic level for the different folding motifs in the irreversible and reversible amyloids derived from a single protein sequence, by combining atomic force microscopy (AFM), cryo-electron microscopy (cryo-EM) and solid-state NMR. We tackle the problem starting from the irreversible lysozyme fibrils, for which the structure could be resolved by cryo-EM; then we study the reversible flexible amyloid variant of the same protein: in this case, the low persistence length and less regular helical twist precludes a high-resolution structure by cryo-EM and so solid-state NMR investigations are performed on the reversible amyloids of human lysozyme. In contrast to the well-ordered irreversible amyloids, the NMR data indicates that the reversible fibrils are relatively heterogeneous, composed of a more-ordered intermolecular β-sheet core surrounded by less-ordered helical

secondary structural elements reminiscent of a molten globule-like state. We conclude that the core of the reversible flexible fibrils, formed during the initial 5 min of unfolding and fibrillization, is likely to consist primarily of the partially unfolded β-sheet region of the globular lysozyme, yielding metastable fibril segments. In contrast, the larger amyloid core identified in the cryo-EM structures of the irreversible fibrils that form after a longer (3 h) unfolding and fibrillization period contain residues from both the β-sheet and alpha-helical sub-regions of the native globular lysozyme, endowing these rigid fibrils with the thermostability characteristics of irreversible amyloids.

## Results and discussion

### Reversible and irreversible amyloid fibrils of lysozyme at the mesoscopic scale

The mesoscopic polymorphism of flexible and rigid fibrils derived from both HEWL and human lysozyme is shown in Fig. 1. In the AFM images (Fig. 1a), the reversible fibrils of HEWL and human lysozyme revealed a morphology resembling semiflexible polymer chains with a diameter of ~2.4 nm (Fig. 1b) and lacking a clear twist periodicity. In contrast, the rigid fibrils of both proteins exhibit an average height of ~7 nm and noticeable periodic height fluctuations along the fibrils. The study on fibril shape (Fig. 1c) indicates that flexible fibrils exhibit a much higher degree of shape fluctuation compared to rigid fibrils. The rigidity within the amyloid core of both reversible and irreversible fibrils is quantitatively evaluated by the persistence length of fibrils through the calculation of their mean square deviation (Fig. 1d) and mean square end-to-end distance (Fig. 1e). Both metrics indicate the extreme flexibility of the reversible fibril core with a persistence length ($L_p$) of tens of nanometers, which is two orders of magnitude lower than the one observed in the rigid irreversible fibrils. This rigidity of amyloid core is validated by the direct observation on the fibrils (Supplementary Fig. 1). Thermal denaturation observed by circular dichroism (CD) spectroscopy indicates that reversible fibrils undergo a disassembly process with a structural transition starting from around 30–40 °C, while the irreversible fibrils maintained their structure until 60 °C (Supplementary Figs. 2–3).

### The atomic resolution structures of human lysozyme and HEWL irreversible fibrils

The well-studied structure of the native human lysozyme[25] (PDB: 7XF6) is made up of two sub-domains: the helical sub-domain A comprising residues 1–40 and 86–130 (with α-helices in residues 5–14, 25–36, 90–100, and 110–115, and $3_{10}$ helices in residues 20–22, 81–85, 105–108, and 122–125) and the β-sheet-containing sub-domain B comprising a short anti-parallel β-sheet with residues 41–85 (with β-strands in residues 43–46, 51–56, and 59–60). The entire structure is stabilized by four disulfide bonds with two bonds in sub-domain A (Cys24-Cys146, Cys48-Cys134), one in sub-domain B (Cys65-Cys81) and one inter-domain disulfide bond (Cys77-Cys95). Disulfide reduction and extended heating at 90 °C is sufficient to fully denature human lysozyme[26], allowing for the formation of heat-irreversible amyloid fibrils. Our cryo-EM analyzes confirm that these fibrils contain a complex 3D structure whose scaffold is the cross-β-sheet motif. Using standard single-particle-based helical reconstruction techniques, we were able to produce an initial de novo 3D model from the 2D classes (Fig. 2a, b) which could then be used for refinement of the irreversible fibril structure of human lysozyme to 2.8 Å resolution (Fig. 2f, Supplementary Fig. 4). The structural model comprises residues 22–34, 50–128 and an additional 12 residues whose sequence could not be identified, covering more than 2/3 of the amino acid sequence of human lysozyme. Apart from a region of tubular density which could not be modeled due to its low quality (lower right in Fig. 2f) the remaining 1/3 of the protein, including the segment 35–40, is not visible in the cryo-EM structure, indicating that these segments are dynamic or heterogeneous in the structure. The amyloid fibril

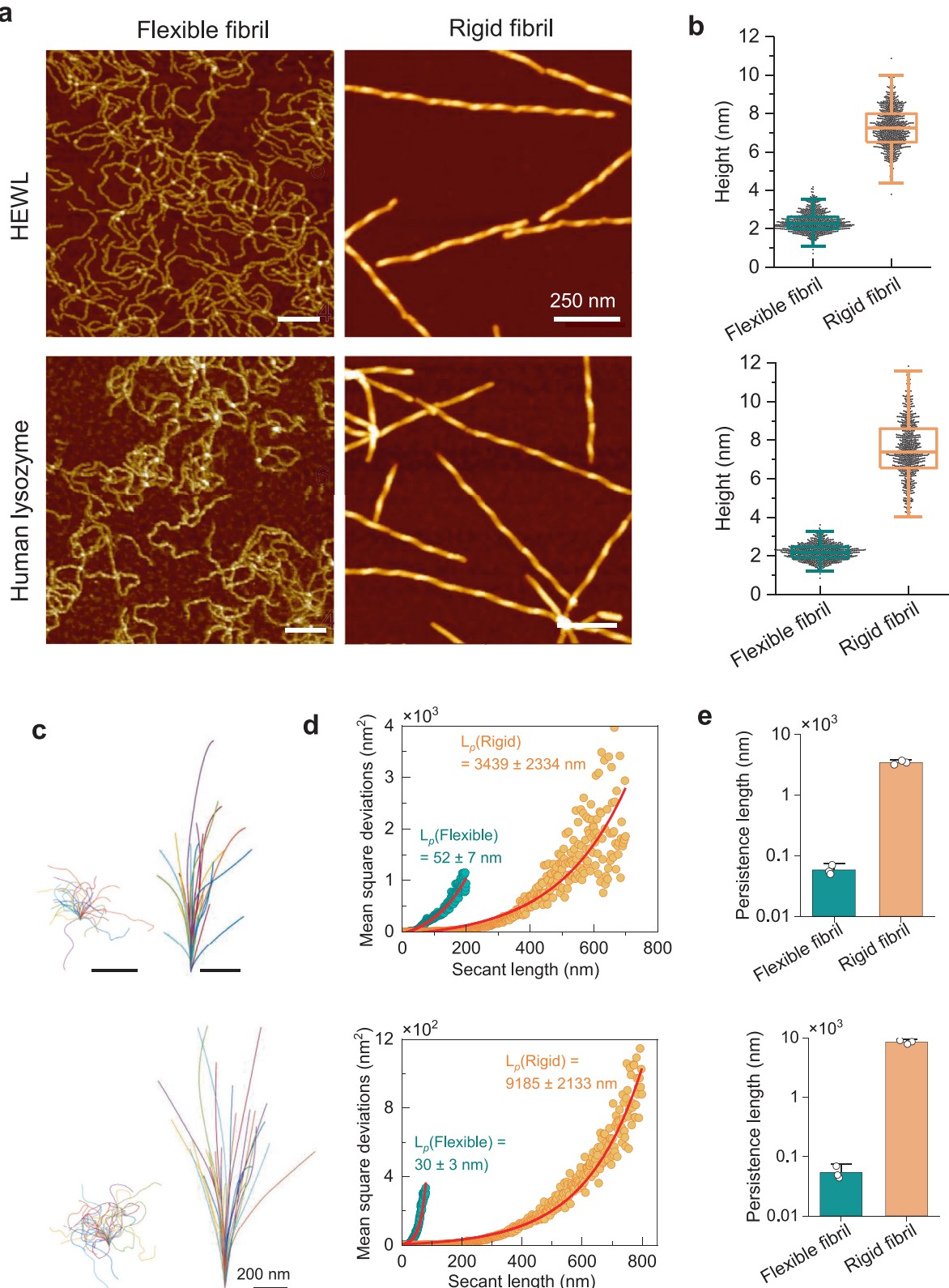

**Fig. 1 | Mesoscopic analysis of reversible and irreversible amyloids of human lysozyme and HEWL.** AFM images (**a**) of flexible (reversible) and rigid (irreversible) fibrils from human lysozyme and HEWL. The height distribution (**b**) and shape fluctuations (**c**) of flexible (left) and rigid (right) fibrils of HEWL (upper) and human (lower) lysozyme. The rigidity of the reversible and irreversible fibrils is measured by the persistence length ($L_p$) via mean square deviation against the secant length[35] (**d**) and confirmed via mean square end-to-end distance[36] (**e**); both indicate the extreme flexibility of reversible fibrils, which is two orders of magnitude lower than that of rigid irreversible fibrils. The upper and lower parts in the panels (**b**–**d**) refer to HEWL and human lysozyme, respectively. More than 700 fibrils from each condition from at least three independent experiments were analyzed in the statistical analysis. $N = 3$ replicates. The box plots in panel b are shown as median value with box range of 25–75% and the outlier whiskers, and the bar plots in panel e are shown as mean values with standard deviation.

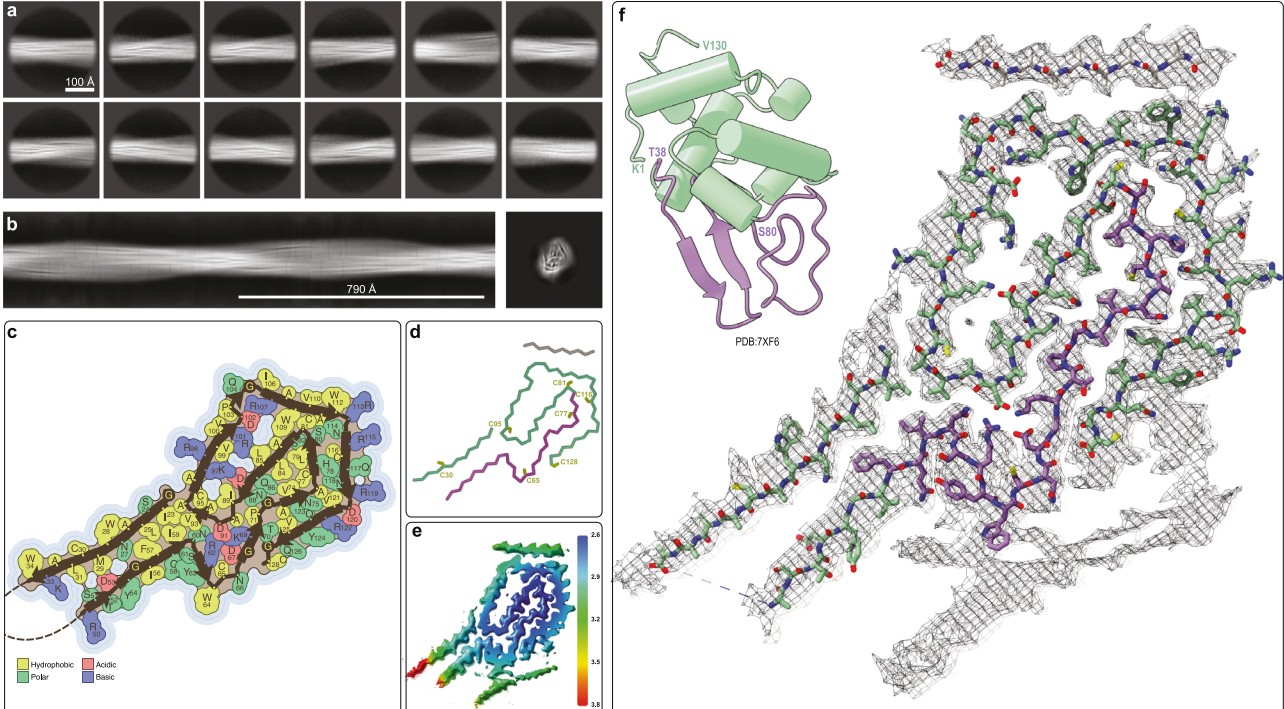

**Fig. 2 | Cryo-EM structure of human lysozyme irreversible amyloid fibrils at 2.8 Å resolution. a** 2D class averages of the irreversible human lysozyme fibrils used to calculate (**b**) an initial de novo 3D model. **c** Topology of the irreversible amyloid fold for human lysozyme showing the location of the β- strands with the sidechains color-coded according to their physiochemical properties. **d** Cα trace of the same amyloid fold color-coded by the secondary structural regions of the native lysozyme fold (α-helical sub-domain A in green and β-sheet sub-domain B in purple) with the 8 cysteine residues labeled (**e**) Cryo-EM map color-coded by local resolution. **f** Cryo-EM density with the atomic model of human lysozyme amyloid fibrils with carbon atoms colored according to the of the sub-domains of the native structure (upper left).

structure is composed of twelve in-register inter-molecular β-sheets (β1: residues 23–32, β2: 51–58, β3: 62–63, β4: 68–69, β5: 74–75, β6: 78–79, β7: 83–87, β8: 96–101, β9: 107–112, β10: 115–119, β11: 120–125 and a 7-residue β-strand of unknown sequence composition). The β-strands are often bordered by glycine residues (i.e., Gly22, Gly68, Gly72, Gly105, Gly127), by prolines (i.e., Pro71, Pro103) or β-arches (Fig. 2). Further typical structural motifs of amyloid fibrils are present and include hydrogen bonded glutamine/asparagine ladders (such as the buried residues Gln86, Gln123, Asn60, Asn75, Asn88, Asn118). In addition, hydrophobic interfaces that include several aromatic residues (Phe57, Trp109, Trp112) are indicated in Fig. 2c. The hydrophobic interface is interspersed with the aforementioned ladders as well as several buried salt bridges (i.e., Arg62-Asp91, Asp67-Lys69, Asp87-Lys97, Arg101-Asp102-Arg107). There are also cavities and a potential solvent exposed salt bridge (i.e., Asp120-Arg122). The surface is rather polar but has a significant number of solvent exposed aromatic residues (i.e., Trp28, Trp34, Tyr54, Tyr124) (Fig. 2c).

As highlighted in Fig. 2d, the cysteine residues are spread throughout the structure without the possibility of forming the native disulfide bonds, rationalizing the necessity of reducing conditions for the formation of these fibrils. Furthermore, a complete unfolding of the native lysozyme is necessary for the irreversible amyloid formation as the sequence elements of both the native helices as well as the β-strands of the anti-parallel β-sheet of the native protein fold are incorporated into the amyloid fibril structure. Interestingly, the residues from the native β-strands are located more on the periphery of the fibril structure (purple in Fig. 2f), while much of the core of the fibril structure is composed of residues from sub-domain B (green). The sub-domain A residues that form helices in the native fold create additional in-register β-sheets in the amyloid core that complement the sub-domain B segments through both polar as well as hydrophobic

interactions such that the two sub-domains of the monomeric structure become intertwined in the fibril structure.

Following a similar strategy, we also solved the structure of the irreversible fibrils of HEWL (Fig. 3f, Supplementary Fig. 5) that were prepared under the same conditions as the human lysozyme fibrils. While the native structures of HEWL (PDB: 7P6M) and human lysozyme (PDB: 7XF6) are very similar, as expected from their 76% sequence similarity, their irreversible fibril structures are very distinct (Figs. 2f and 3f). Neither their overall folds, the sequence locations of the β-strands, nor the core hydrophobic and charged interactions are similar. Considering the divergence in their sequences and that evolutionary selection did not occur under the boundary condition of preserving this fibril fold, this result is not entirely surprising. In details, the HEWL amyloid fibril structure comprises residues 26-101 and is composed of 11 inter-molecular in-register β-sheets (β1: residues 28-30, β2: 33-38, β3: 43-45, β4: 51-52, β5: 55-62, β6: 73-76, β7: 80-83, β8: 89-92, β9: 93-98 and two β-strands, of 6 and 4 residues with unknown sequence composition). The β-strands are sometimes bordered by Gly residues (i.e., Gly26, Gly49, Gly54, Gly71, Gly102), by prolines (i.e., Pro79) or β-arches (Fig. 3c). There are asparagine ladders (such as the buried Asn37, Asn59, Asn74, and Asn77 [which is buried because there is another undefined β-strand close by as highlighted in Fig. 3f]). Two hydrophobic interfaces, including a peculiar Trp triad (Trp28, Trp62, Trp63) are surrounded with the aforementioned ladders, several buried salt bridges (such as Glu35-Arg45, Asp52-Lys33, Asp66-Arg61, Asp48-Lys97) with nearby cavities, as well as other polar residues including serine and threonine (Fig. 3c). The surface is rather polar, but has one solvent-exposed aromatic residue (i.e., Phe38, please note other apparent solvent-exposed hydrophobic residues shown in Fig. 3c such as Phe34, Val29, Trp63, Leu75, Leu83 and Leu84 are not considered solvent exposed as there are unresolved densities surrounding

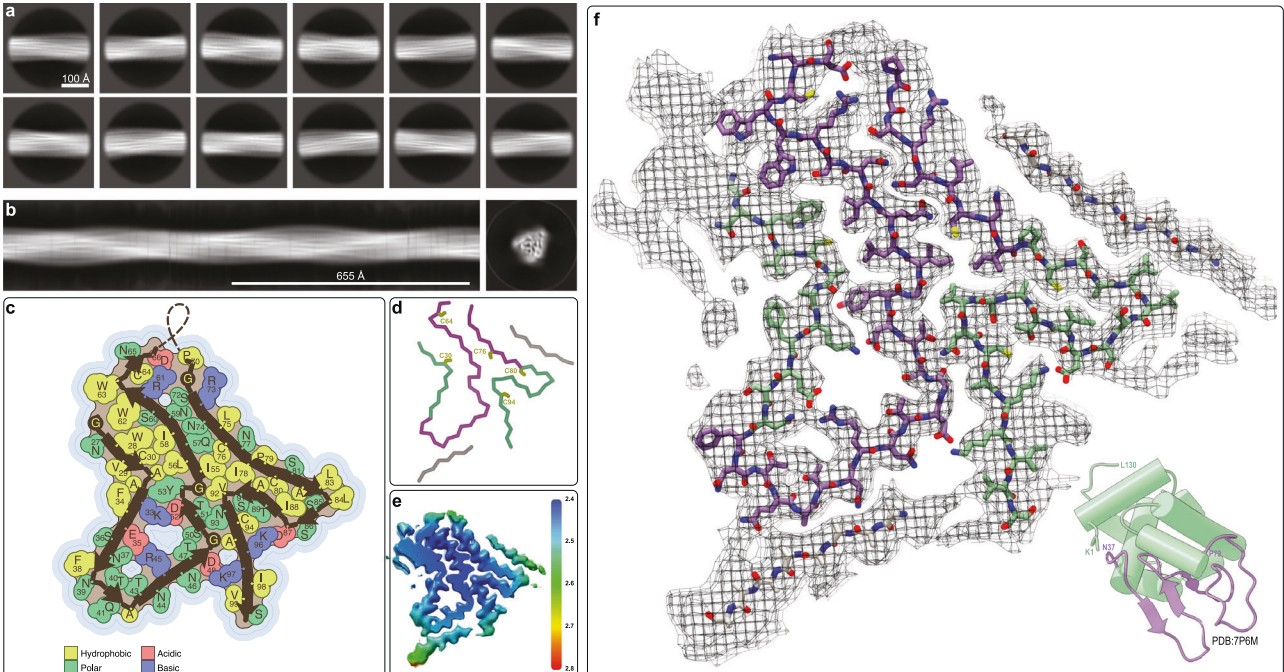

**Fig. 3 | Cryo-EM structure of HEWL irreversible amyloid fibrils at 3.2 Å resolution. a** 2D class averages of the irreversible HEWL fibrils used to calculate (**b**) an initial de novo 3D model. **c** Topology of the irreversible amyloid fold for HEWL showing the location of the β-strands with the sidechains color-coded according to their physiochemical properties. **d** $C_\alpha$ trace of the same amyloid fold color-coded by the secondary structural regions of the native lysozyme fold (α-helical sub-domain A in green and β-sheet sub-domain B in purple) with the 8 cysteine residues labelled. **e** Cryo-EM map color-coded by local resolution. **f** Cryo-EM density with the atomic model of HEWL with carbon atoms colored according to the sub-domains of the native structure (lower right).

them as shown in Fig. 3f) and there is also a partially solvent exposed salt bridge (i.e., Asp87-Lys96).

While the 3D structure of the irreversible HEWL amyloid fibril structure is largely distinct from its human analog both at the secondary structure level as well as the side chain interface between the secondary structures, once again the two sub-domains of the native fold are intertwined in the amyloid structure with the anti-parallel β-sheet segment of sub-domain B forming partially the core of the fibril structure surrounded and complemented by segments from the helical sub-domain A of the native fold.

### Reversible fibrils of human lysozyme are heterogeneous and dynamic

Having established that two distant homologs that share a common native fold form very distinct irreversible amyloids, we wanted to compare these to the reversible amyloids that form during shorter denaturation periods. After a short heat denaturation at 80–90 °C under reducing conditions, the reversible fibrils of both HEWL and human lysozyme then form close to room temperature and unlike the irreversible fibrils, they unfold/dissolve at a temperature above ~45 °C[24] (Supplementary Fig. 2). Based on atomic force microscopy (AFM) data (Fig. 1) we suggested that the flexible reversible fibrils with their relatively short persistence length are composed of a lower level of structural organization than the rigid irreversible amyloid fibrils. Cryo-EM image analysis is consistent with this interpretation but also indicated that high-resolution structural information could not be obtained with this technique. Our attempts to resolve any structural detail from the 2D classes were unsuccessful despite trying out a range of box sizes and shorter inter-box distances in an attempt to account for the short persistence length of the filaments (Supplementary Fig. 6). Hence, solid-state NMR structural studies were undertaken on human lysozyme for which stable isotope labelled recombinant protein could be produced. Figure 4a shows a 2D [$^{13}$C,$^{13}$C]-DARR spectrum of reversible fibrils of $^{13}$C,$^{15}$N-labeled human lysozyme measured on an

850 MHz $^1$H NMR spectrometer. The presence of very broad cross peaks with an overall poor spectral resolution is striking since it is atypical for a well-ordered cross-β-sheet motif of an amyloid fibril. A superposition with a simulated 2D spectrum based on the structure of the irreversible fibrils and using typical line widths for fibrils highlights the extent of line-broadening of the measured spectra (Fig. 4a). Also, the presence of several polymorphs in the sample can not account for the broad peaks because they usually yield multiple distinct cross peaks per $^{13}$C-$^{13}$C moiety[27]. In addition, the low level of superposition by visual inspection demonstrates that the reversible fibrils are distinct in structure from the irreversible fibrils. The approximately 750 Hz line width of the cross peaks is due to both microsecond to millisecond dynamics contributing approximately 250 Hz to the line width via transverse relaxation (Fig. 4c) and structural heterogeneity contributing approximately 500 Hz to the line width (Fig. 4d). A chemical shift distribution of 500 Hz corresponds to 2.3 ppm at 850 MHz $^1$H, sufficient resolution to distinguish between a particular $^{13}$C moiety existing in a helical or β-sheet secondary structure element. However, with a 2.3 ppm line width there would be some overlap/uncertainty distinguishing between helical or sheet and less well-ordered structures including random coil-like configurations. Nonetheless, for Ala residues, distinct cross peaks exist in both the alpha-helical and β-sheet chemical shift regions of the spectrum (Fig. 4a) providing some sequence restraints on the proposed model discussed in the next section. To characterize the topology of the β-sheet, we measured a 2D PAIN experiment on a sample that contained a co-fibrillized mixture of $^{15}$N-labeled protein and $^{13}$C-labeled protein (Fig. 4b). A cross peak between $^{15}$N and $^{13}$C in the spectrum thereby demonstrates the presence of close inter-molecular contact typically observed in a β-sheet in which each strand is from a different protein chain. Counting the broad cross peaks between $^{15}$NH and $^{13}$C$_\alpha$ pairs suggests that approximately 15–25 residues are involved in an inter-molecular β-sheet conformation. Furthermore, the broad cross peaks between $^{15}$NH and $^{13}$C$_\beta$ of alanine indicates that approximately five Ala

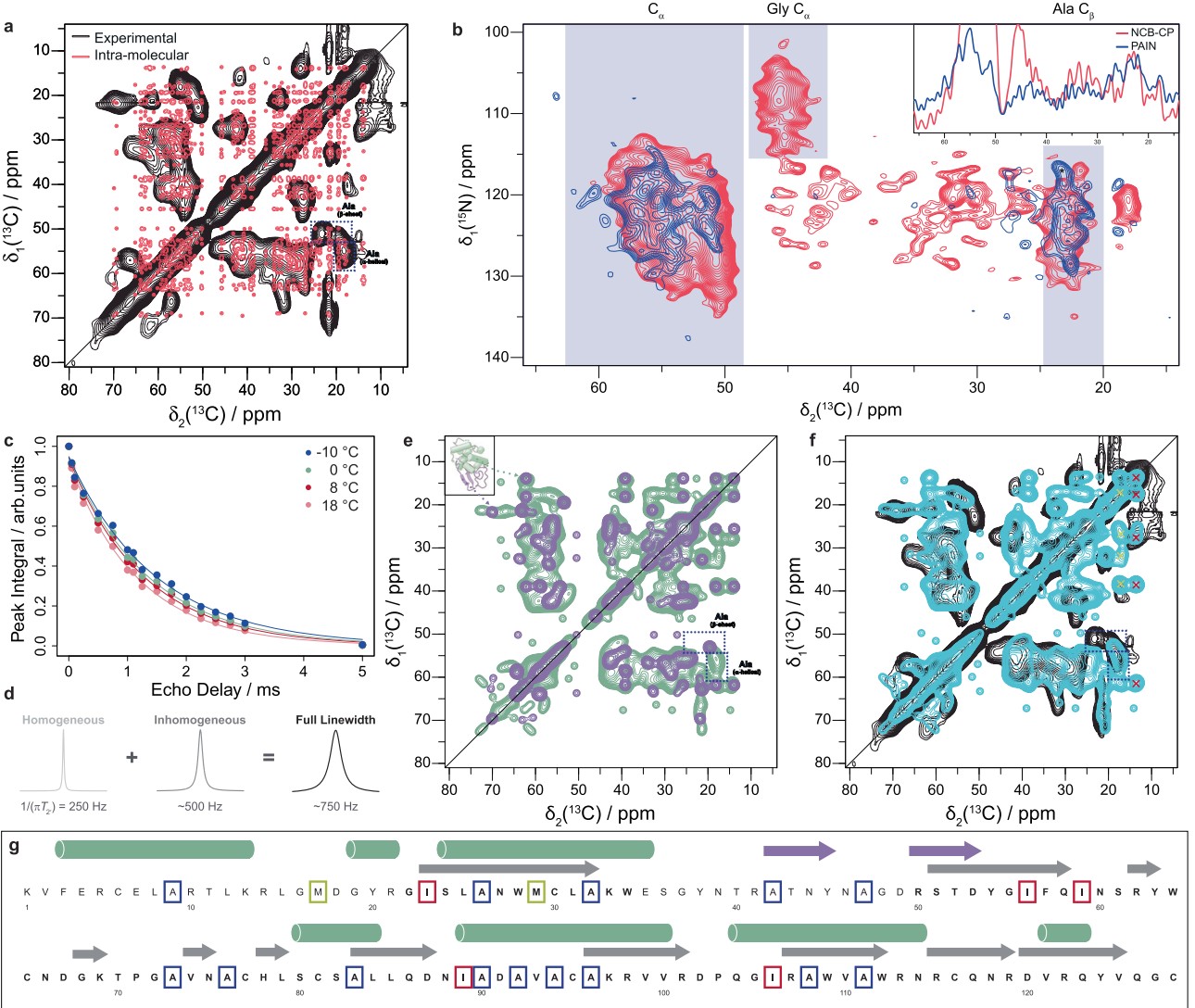

**Fig. 4 | Solid-state NMR spectra and analysis of reversible fibrils of human lysozyme demonstrating the formation of dynamic heterogenous inter-molecular β-sheets as well as helical molten globule-like structures. a** Aliphatic region of a 2D [$^{13}$C, $^{13}$C]-DARR spectrum in black (Table S1) overlaid with a simulated spectrum in red of the irreversible fibrils using typical linewidths for an amyloid structure. Both helical and beta-structured Ala sidechain peaks are highlighted in the spectra. **b** Detection of intermolecular contacts in reversible lysozyme fibrils between a $^{15}$N-labeled protein and a $^{13}$C-labeled protein by a 2D PAIN experiment yielding a $^{15}$N-$^{13}$C correlation spectrum (blue). The spectrum is overlaid with a 2D NC-CP (red) spectrum. The shaded regions show typical $^{13}$C chemical shift ranges for the indicated atom types (the * symbol indicates signals for an out-of-register inter-molecular contact). The inset shows cross-sections used to adjust the contour lines of the 2D spectra. **c** Measurement of bulk $^{13}$C T2′ relaxation times at different temperatures by a variable-delay spin echo and fit by a single exponential (Table S2). **d** T2′ relaxation measurements indicating that the observed 750 Hz peak width in the 2D [$^{13}$C, $^{13}$C]-DARR spectrum at 8 °C (Table S1) is attributed to 250 Hz from dynamics and 500 Hz from structural heterogeneity. **e** Simulated 2D [$^{13}$C, $^{13}$C]-DARR spectra of the sub-domain A residues from the native human lysozyme structure in green overlaid with that of the sub-domain B residues 37–80 in the irreversible fibril structure in purple (both spectra with a peak width of 750 Hz to match the experimental data). Ala sidechain peaks are highlighted as in (**a**). **f** An overlay of the experimental spectrum (black) from (**a**) with a sum of the two simulated spectra (cyan) from (**e**). **g** The primary sequence of human lysozyme showing the secondary structural elements of both the soluble monomer (purple and green) and the irreversible fibril (gray). The letters in bold indicate residues in the model of the irreversible fibril. Some residues whose absence or presence are discussed in the text are colored to highlight their positions: Met (yellow), Ile (red) and Ala (blue).

residues are part of the β-sheet conformation. Usually, a superposition between the PAIN spectrum (blue) and an NC-CP spectrum (purple) shown in Fig. 4b can be used to distinguish between in-register or out-of-register inter-molecular β-sheets, the former being parallel by definition and the latter including both parallel and anti-parallel arrangements[27]. However, the low resolution of the spectra precludes a straightforward interpretation. The overall superposition in the region that correlates nitrogen with alpha carbon resonances (Fig. 4b, indicated area between 48–60 ppm) may be interpreted to support an in-register β-sheet content, while the presence of one intra-molecular $^{15}$N-$^{13}$C$_\beta$-cross peak in the PAIN that is absent in the NCB spectrum may

indicate the presence of out-of-register inter-molecular β-sheets which can include anti-parallel β-sheets. Indeed, the presence of anti-parallel and parallel β-sheet secondary structure was proposed for the rever-sible fibrils based on our previous FT-IR and CD analyzes[24].

## Constructing a plausible 3D model of the reversible fibrils of human lysozyme

The low-resolution data on the reversible fibrils indicate that they are composed of both helical and inter-molecular β-sheet secondary structures that contain alanine residues (Fig. 4a–g) in both the helices and ca. 5 in the ca. 15–25 residues that comprise the inter-molecular β-

strands. This is in line with CD investigation (Supplementary Figs. 2–3), where the reversible fibrils show the mixed structure of helical and β-sheet conformation. Furthermore, there is a large structural heterogeneity and the presence of dynamics typically observed in molten globule-like states[28,29]. Taking into account that the reversible fibrils are formed after annealing at 80 °C and dissolve above ~40 °C, as well as the published work on lysozyme heat denaturation under acidic, but non-reducing conditions by Dobson and coworkers[30], we can begin to piece together a reasonable model for the formation of the reversible fibrils and their conversion to irreversible fibrils. Thermal treatment of lysozyme was shown to involve[30] "a cooperative quick loss of native tertiary structure, followed by progressive unfolding of a compact, molten globule-like state ensemble as the temperature is increased" with sub-domain B comprising the anti-parallel β-sheet unfolding first at around 25–30 °C and the helical secondary structure of sub-domain A only above 30–40 °C or even higher. The short heat treatment for reversible lysozyme fibrils formation may only lead to a partial denaturation of lysozyme with the loss of native tertiary structure and different degrees of loss of secondary structure. It is thus plausible that the (majority of the) helices of sub-domain A are at least partially preserved at room temperature after heat denaturation thereby hindering the formation of the irreversible fibril fold for which they would need to adopt a β-structure. The reversible fibrils (formed near room temperature) must then include these helical secondary structures in a molten globule-like state, yielding the observed broad lines in the solid-state NMR spectra, with, nonetheless, their inter-molecular β-sheets composed of 15–25 residues of the sub-domain B (including ca. 5 alanine residues, Fig. 4g) capable to form due to their lower unfolding temperature. This is also in line with the CD spectrum of reversible fibrils (Supplementary Fig. 2–3) that indicates a mixture of helical and β-sheet structure, with no significant difference compared to native lysozyme[31]. Furthermore, the sheets are of a heterogeneous nature as requested by their solid-state NMR spectra and their short persistence length measured by AFM analysis. To test these models, we calculated simulated 2D DARR (Fig. 4 and Supplementary Fig. 7) and 2D NCB spectra for the native human lysozyme (Supplementary Fig. 8b), the irreversible fibrils, and a chimeric structure consisting of sub-domain A of the native fold and the region of the irreversible fibrils comprising the residues from sub-domain B of the native fold (Fig. 4f). It is evident that neither the irreversible fibril structure alone (Supplementary Fig. 7c), nor the native structure (Supplementary Fig. 7b) can fully explain the solid-state NMR spectra. In contrast, and in support of the proposed model, the simulated spectra of the hypothetical chimera structure resemble the experimentally measured NMR spectra reasonably well (Fig. 4f and Supplementary Fig. 8c), for example the position of the Ala cross peaks in the 2D DARR discussed above and indicated in Fig. 4. It is also noted that some predicted cross peaks for Met and Ile (highlighted by yellow and red crosses, respectively in Fig. 4e) are absent in the measured spectra while the corresponding diagonals are observed. This loss of expected signals indicates the presence of intermediate dynamics that further hint at the overall dynamics of the helical segments, as both Met and Ile are mostly localized within subdomain A.

## Reversible and irreversible fibrils

It has been suggested that the complete denaturation of almost any protein can lead to the formation of highly stable amyloid fibrils as almost every protein has amyloidogenic sequence segments[32]. Based on our structural studies on reversible and irreversible fibrils of lysozyme the folding energy landscape is not only rough with multiple minima, but also changes with time when the system is taken out of equilibrium. We suggest that this type of pathway-dependent protein folding/amyloid polymorphism may be generally applicable to proteins, particularly for those that contain relatively small regions/sub-domains that are able to form inter-molecular β-sheets while retaining

a significant portion of helical secondary structures that dynamically interact with each other, as typically observed in molten globule-like states. This phenomenon was also previously observed in $\beta_2$-microglobulin fibrillization[33], in which the distinct competing on/off pathways played a key role in the assembly mechanism, leading to the formation of different kinetically trapped species of fibrillar assemblies. In the present case, a short heat-treatment under reducing conditions induces the loss of lysozyme tertiary structure and then secondary structure in sub-domain B, with different degrees of unfolding in sub-domain A. The subsequent annealing promotes the formation of reversible fibrils with heterogeneous sheets and heterogeneous intact helical structures in sub-domain A, resulting in very broad NMR resonances. Such a reversible fibril could in principle fold back into its native state or partially unfolded state. In contrast, under the longer heat-treatment in the presence of a reducing agent, native lysozyme undergoes a complete unfolding, which allows it to fold into irreversible fibrils, with twelve and ten in-register inter-molecular β-sheets for human lysozyme and HEWL, respectively. Interestingly, the conversion from reversible fibrils into irreversible fibrils of full-length lysozyme[24] (Supplementary Fig. 9) is possible with extended heat denaturation, during which the β-sheets re-organize and the helical secondary structures are lost, allowing irreversible fibrils to incorporate these latter sequence segments into their in-register β-sheet fibril core. This statement is validated through direct AFM observation (Supplementary Fig. 10), which captures the transformation of reversible fibrils into irreversible fibrils. It is thereby noted that a kinetic trapping dictated by local helical secondary structure formation appears to be the key for reversible versus irreversible fibril formation (Fig. 5) in a selection process that may be generally applicable to many other proteins including functional reversible/irreversible amyloid systems.

## Methods

### Materials

The commercial hen egg white lysozyme (HEWL, Sigma Aldrich L6876), human lysozyme (Sigma Aldrich L1667) and 1,4-Dithiothreitol (DTT, D0632) and other chemicals were purchased from Sigma Aldrich. The batch L1667 from Sigma was found to contain a minute amount of rice protein contaminant, i.e., trypsin inhibitor, a residue of protein expression. Trypsin inhibitor was also found to form amyloid fibrils, its structure was solved by cryoEM and will be presented in a separate study. The isotopically labeled recombinant human lysozyme was expressed and purified as described below.

### Protein expression and purification

Recombinant human lysozyme was expressed from Pichia pastoris and purified following an optimized version of a published protocol[34]. In brief, yeast cells were grown in buffered minimal glycerol media (BMG; 100 mM potassium phosphate pH 6.0, 1.34 % yeast nitrogen base, 1% glycerol, $4 \times 10^{-5}$ % biotin) for 48 h at 30 °C. The cultures were diluted 1:50 in fresh BMG media and allowed to grow for an additional 48 h. The cells were harvested by centrifugation (4 °C, 5000 g, 5 min), re-suspended in buffered minimal methanol media (BMM; 100 mM potassium phosphate pH 6.0, 1.34 % yeast nitrogen base, 0.5 % methanol, $4 \times 10^{-5}$ % biotin) and grown at 23 °C for 96 h. During the induction period, additional 0.5 % methanol was added after 24 h and 48 h. Cultures were centrifuged (4 °C, 9000 g, 10 min), pellets were discarded, and the supernatant was centrifuged a second time (4 °C, 9000 g, 10 min) and filtered (0.45 μm filter). Lysozyme purification was performed by cation exchange chromatography using a HiTrap SP HP column following manufacturer's instructions. Lysozyme was eluted by a linear NaCl gradient, and fractions were collected and checked by SDS-PAGE followed by Coomassie blue staining. Pure fractions were pooled, dialyzed against Milli-Q water for 3 days with daily two water exchanges, and then stored at −80 °C. Lysozyme activity was assessed

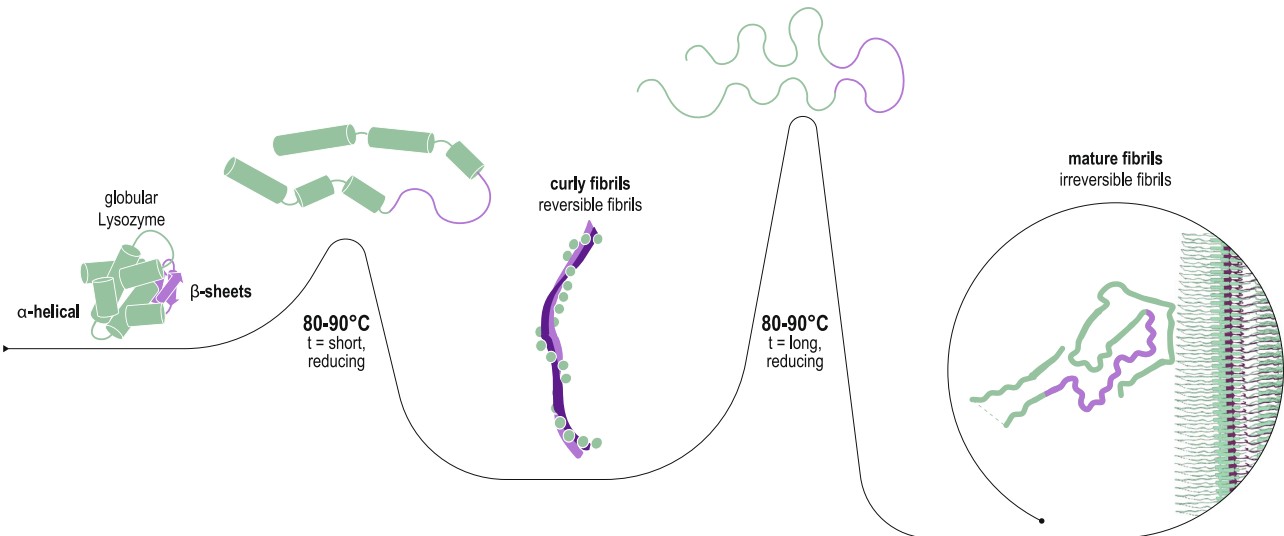

**Fig. 5 | Proposed mechanism of temperature-induced unfolding and transitioning from reversible to irreversible lysozyme fibrils.** The sub-domain B (purple) of the native fold is the first region to denature upon heating and can form the core of the reversible fibrils upon cooling. Longer heating time leads to a full denaturation of the protein, allowing the sub-domain A residues (green) to adopt the beta structure required for the formation of the irreversible fibrils.

by incubation with *Micrococcus lysodeikticus* cells. When needed, aliquots were thawed on ice, concentrated by centrifugation and the pH adjusted to 7.0.

### Amyloid fibril formation

For commercial human lysozyme, the protein powder (Sigma Aldrich L1667) was dissolved in Milli-Q water at 30 mg/ml at room temperature for 3 hours or 4 °C overnight, and then filtrated with 0.45 μm filter. Freshly prepared DTT (1 M) and NaCl solutions (1 M) were added to reach a final protein solution which was adjusted to pH 7 prior to incubation. The flexible fibrils were prepared by incubating the solution (20 mg/ml lysozyme, 20 or 100 mM DTT, 10 mM NaCl) for 5 min at 85 °C with a magnetic stirring at 300 rpm. The rigid fibrils (20 mg/ml lysozyme, 100 mM DTT, 10 mM NaCl) were prepared by incubating the solution for 3 h at 85 °C with a magnetic stirring at 300 rpm. For commercial HEWL lysozyme, the protein powder (Sigma Aldrich, L6876) was dissolved in Milli-Q water at 30 mg/ml at room temperature for 3 hours or 4 °C overnight, and then filtrated with 0.45 μm filter. Freshly prepared DTT (1 M) and NaCl solutions (1 M) were added to reach a final protein solution which was adjusted to pH 7 prior to incubation. The flexible fibrils were prepared by incubating the solution (20 mg/ml lysozyme, 20 or 100 mM DTT, 10 mM NaCl) for 5 min at 90 °C with a magnetic stirring at 300 rpm. The rigid fibrils (20 mg/ml lysozyme, 100 mM DTT, 10 mM NaCl) were prepared by incubating the solution for 3 hours at 90 °C with a magnetic stirring at 300 rpm. Immediately after incubation, the sample was cooled in an ice bath for 30 min and stored at 4 °C (see Table S3 for details). For the expressed human lysozyme reversible fibrils used in solid-state NMR, the protein was purified as described above with either $^{15}$N,$^{13}$C double-labeled lysozyme or a 1:1 mixture of $^{15}$N and $^{13}$C single-labeled human lysozyme. Human lysozyme (20 mg/ml in Milli-Q water, pH 7) was mixed with 10 mM NaCl and 20 mM DTT, and a 250 μl aliquot was incubated in an oil bath with a magnetic stirring at 300 rpm for 5 min at 80 °C. Samples were allowed to slowly cool to room temperature, before they were stored at 4 °C until further use.

### AFM and data analysis

Aliquots of lysozyme reversible and irreversible fibril solutions were diluted to 0.1 mg/ml with Milli-Q water. Immediately, an aliquot (10 μl) of diluted protein solution was deposited on a freshly cleaved mica, incubated for 2 min, rinsed with Milli-Q water and dried by a gentle

flow of nitrogen gas. AFM measurements were carried out by a Bruker multimode 8 scanning probe microscope (Bruker, USA) with an acoustic hood to minimize vibrational noise. AFM imaging was operated in soft tapping mode under ambient conditions, using a commercial silicon nitride cantilever (Bruker, USA) at a vibration frequency of 150 kHz, and a relatively soft tip-sample interaction was applied. AFM images were flattened using Nanoscope 8.1 software (Bruker, USA), and no further image processing was applied. Analysis of fibril shape fluctuations was initially assessed following the literature[35] and then a more extensive statistical analysis, including height and persistence length was carried out using the FiberApp software[36] on flattened AFM images, and persistence length was calculated by fitting the mean-squared end-to-end distance between contour segments. More than 700 fibrillar aggregates for each type of fibril collected from at least three independent experiments were analyzed in the AFM statistical analysis.

### Electron microscopy grid preparation and data collection

Cu R2/2 300 mesh grids (Quantifoils) were glow discharged at 25 mA for 30 seconds. Freshly glow-discharged grids were used in a Vitrobot Mark IV (Thermo Fisher Scientific) with its chamber set at 100% humidity and at a temperature of 15 °C. Fibrils (4 μl aliquots) were applied to the grid and blotted for 5 seconds using a blotforce of 1 after a 5 second wait time, and subsequently plunge-frozen into a liquid Ethane/Propane mix. The grids were clipped and immediately used or stored in liquid nitrogen.

Data acquisition was performed on a Titan Krios (Thermo Fisher Scientific) operating at 300 kV equipped with a Gatan Imaging Filter (GIF) with a 20 eV energy slit using Gatan's K3 direct electron detector in counting mode. Movies of 40 frames were collected using EPU software (Thermo Fisher Scientific) at a magnification of 130kx and a dose rate of approximately 8 e/pix/sec and total dose of ca. 63 e/Å$^2$ (see Table S5 for details) in counted super-resolution mode at a pixel size of 0.325 Å and binned to 0.65 Å.

### Image processing

Image processing and helical reconstruction were carried out with RELION 4.0, following the procedure for amyloid structures described by Scheres[37]. RELION's own implementation of MotionCorr2[38] was used to correct for drift and dose-weighting and CTF estimation was done using Ctffind4.1[39]. Individual filaments were manually selected,

and segments were extracted in a 333 Å box with an inter-box distance of 33 Å. For 2D classification, the extracted segments were binned 4x to a pixel size of 2.6 Å. For both human and hen lysozyme fibrils, there were significant amounts of straight and twisted classes. The classes that belonged to the twisted polymorph were selected, cropped to a 90-pixel box and input into *relion_helix_inimodel2d* in order generate an initial 3D model (Figs. 2b and 3b). The starting volumes were generated with a left-handed twist to match what had been observed in the AFM analyzes and then used for the subsequent 3D refinement steps. At this stage, the crossover distances are only approximate because this parameter is not well-defined in the process of stitching together the class averages into a full filament.

### Model building and refinement

Alignments of the segments from the selected 2D classes were performed against the helix_inimodel2d model in 3D auto refine jobs until the resolution of these refinements reached the Nyquist limit. These aligned segments and the model from the 3D refinement were used in a 3D classification with 2 classes to select for the segments that fit best to the refined model. The selected segments were re-extracted at 1.3 Å per pixel and further refinements were performed until no further improvement could be made. This was followed by iterative rounds of CTF refinement, motion correction refinement (Bayesian polishing) and 3D refinements. In the case of the HEWL reconstruction, the final rounds of refinement included an input mask with a 7 Å soft edge and using solvent flattened FSCs. For both filament reconstructions, such mask was used in a final postprocessing step. ModelAngelo[40] was used to create an initial model. The human map could be fit without providing an input sequence such that the output sequences from ModelAngelo were accurate enough that when used as a query with BLASTp they returned human lysozyme as the top hit. For the lower resolution HEWL maps, ModelAngelo required the protein sequence as input in order to get a good fit. Still, attempting to fit the human sequence into the density for HEWL yielded a very poor fit. The output models from ModelAngelo were manually adjusted and extended in COOT[41] followed by real-space refinement as a 9-layer fibril in ISOLDE[42] with symmetry and secondary structure restraints. A final real-space refinement was done with Phenix[43] in order to get a reasonable estimate of the atomic B-factors. The outer four layers, which often diverge slightly in structure due to their placement at the edges of the model of the model were removed for deposition of the central 5-layers in the PDB. Figures were prepared with CCP4MG[44] and UCSF Chimera X[45] and Amyloid Illustrator. The latter uses a different algorithm to define the location of beta-strands and so there slight mismatches between its output and the secondary structure that we define in the main text and in Fig. 4g using STRIDE[46]. We chose not to use the DSSP[47] secondary structure definitions because it fails to locate some unambiguous beta-strands.

### Solid-state NMR

The various fibrils were sedimented directly into a 3.2 mm MAS rotor in an ultracentrifuge at 210'000 g and 4 °C overnight using home-built tools[48]. Solid-state NMR experiments were performed at 20.0 T static magnetic field (850 MHz ${}^1$H Larmor frequency) using a triple-resonance 3.2 mm E-free probe (Bruker) at an MAS frequency of 17 kHz. Spectra were processed with TopSpin version 3.6 (Bruker) and referenced to adamantane as an external standard. 2D spectra were processed with a shifted cosine apodization function (SSB = 2.3 or 2.5) and automated baseline correction. The sample temperature was estimated via the water line and adjusted to 8 °C[48]. For temperature-dependent measurements, the gas flow and target temperature were adjusted such as to reach a sample temperature of 18, 8, 0 and −10 °C. We used standard solid-state NMR pulse sequences for the PAIN and DARR experiments[49,50].

### NMR Spectra Simulations

The DARR and NCA spectra were simulated in an empirical manner based on the cryo-EM structure of the fibrils of the human lysozyme protein as well as the crystal structure of the native monomeric protein. The chemical shifts of the individual atoms from both the fibril and the native structures were predicted by SHIFTX2[51]. Based on the structures and the predicted chemical shifts, the diagonal- and cross-peak positions were predicted based on an interatomic distance cutoff of 5 Angstrom using the algorithm FLYA[52]. The peaks of the resulting peak list were than filtered according to the information which needed to be extracted, e.g. in some simulations only the peaks originating from intra-residual interactions were taken into account. The filtered list of peaks was then passed to a Python script (https://github.com/lucawenchel/gaussian_spectrum) to calculate the spectra by applying a 2D gaussian functions with the standard deviation chosen as to produce peaks with the desired peak width.

### Reporting summary

Further information on research design is available in the Nature Portfolio Reporting Summary linked to this article.

## Data availability

The coordinates for the hen egg white and human lysozyme fibril models have been deposited in the PDB with accession codes 8QV8 and 8QUT, respectively. Likewise, the Cryo-EM maps for hen egg white and human lysozyme fibrils have been deposited in the EMDB with accession codes EMD-18669 and EMD-18663, respectively. PDB codes of previously published structures used in this study are 7XF6 and 7P6M. All the other relevant data support the findings are available within the paper, the supplementary material files and source data. The source data are provided as Source Data file with this paper. Source data are provided with this paper.

## Code availability

The unpublished Python script for simulating PAIN and DARR spectra and developed for this work can be accessed at https://zenodo.org/records/13769801.

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

## Acknowledgements

We would like to thank the staff at the ScopeM ETHZ for assistance with the cryo-EM data-collection analysis. M.W. acknowledges a PhD stipend by the Günthard-Stipendium für Physikalische Chemie.

## Author contributions

R.M. designed the project. G.C. and J.Z. purified the single and double-labeled lysozyme protein. J.Z. and G.C. performed fibrillization. J.Z. performed mesoscopic analysis. L.F. J.G., D.R., A.P., and R.R. conducted the cryo-EM and data analysis. M.W., B.H.M., H.K., L.W., and R.R. performed the ssNMR measurements and analyses. J.Z., L.F., G.C., Y.C., M.P, J.G., R.R., and R.M. wrote the manuscript, and all authors contributed to the correction and editing of the manuscript.

## Funding

## Competing interests

The authors declare no competing interests.
