## [Peer Review File · Nature Communications]

A structural rationale for reversible vs irreversible amyloid fibril formation from a single proteinREVIEWER COMMENTS

Reviewer #1 (Remarks to the Author):

This study presents the argument that a single amino acid sequence, under varied degrees of denaturation, can generate distinct amyloid-like fibrils. The data presented suggest that these fibrils differ in overall morphology, secondary structure content and local dynamics.

The model protein, lysozyme, used in this article is a globular protein seen ubiquitously as a standard throughout structural molecular biology. In this study lysozyme is used to understand the energetic landscape by which proteins assemble into stable and often irreversible supramolecular complexes.

The authors present structures of human and HEWL lysozyme fibrils derived from cryo-EM measurements. These were prepared from monomers exposed to “long” heat denaturation and due to the fiber’s resistance to subsequent heat denaturation and rigid homogenous core amenable to cryo-EM investigation, they are referred to as “irreversible” fibrils. The authors also provide SSNMR data for “reversible” fibrils which are classified as such due to their lack of rigidity and sensitivity to heat denaturation. By comparing regions present in the native globular state with those seen in both reversible and irreversible fibrils, the authors conclude that partial denaturation maintains helical structure elements incapable of contributing to canonical beta-sheet fibril formation and that complete denaturation is required for extensive beta-sheet packing present in irreversible fibrils.

My comments on the manuscript are as follows:

1) Regarding presentation of cryo-EM data, I note/recommend the following: (1) A density map displaying a half pitch length along the fibrillar axis would be helpful to include and would allow the reader to visualize the degree of resolution achieved along the fibrillar axis; (2) at least one sample micrograph should be included in the supporting material to allow the reader to visualize the quality of the raw data; (3) For figures 2 and 3, the EM data presented in (a) and (b) does not include a scale bar making visual analysis impossible. With twist angles of 1.22 and 1.00, expected half-pitches would be 705.2 Å and 860.4 Å, yet (b) in both figures appear to be the same length. (4) the quality of the 2D classes appear suboptimal when considering the the final resolution.

2) Cryo-EM data and processing – In table S4 the pixel size is labeled 1.3 Å, but on page 18 line 14, it says “binned 4x to a pixel size of 2.6 Å”. It was also not mentioned if this was acquired in super resolution mode, which may account for this discrepancy. In the same table there appears to be a marked improvement in the human lysozyme and not in the HEWL with no explanation.

The presence of excess electron density and unidentifiable regions of seemingly beta-sheet structures seen in the cryo-EM maps, makes it difficult to assess the quality of the structures. The conclusions rest on the foundation of regional comparison between reversible and irreversible

structures, yet both the cryo-EM and SSNMR results seem to be somewhat ambiguous. With the presence of additional unknown electron density, how was it determined what regions belong to rigid structured portion? Could this have been verified with SSNMR? I could not access the EMDB or PDB entries to further assess this, but by inspection, it is difficult to assess how the stated resolution is supported by the data. It would be helpful to include FSC curves in the supporting material. The following provide examples of fibril structures solved to 2.8-3.4 Å resolution:

Wang, Li Qiang, Kun Zhao, Han Ye Yuan, Qiang Wang, Zeyuan Guan, Jing Tao, Xiang Ning Li, et al. 2020. "Cryo-EM Structure of an Amyloid Fibril Formed by Full-Length Human Prion Protein." *Nature Structural & Molecular Biology* 27 (6): 598–602.

Guerrero-Ferreira, Ricardo, Nicholas Mi Taylor, Daniel Mona, Philippe Ringler, Matthias E. Lauer, Roland Riek, Markus Britschgi, and Henning Stahlberg. 2018. "Cryo-EM Structure of Alpha-Synuclein Fibrils." *eLife* 7 (July). <https://doi.org/10.7554/eLife.36402>.

Cao, Qin, David R. Boyer, Michael R. Sawaya, Peng Ge, and David S. Eisenberg. 2019. "Cryo-EM Structures of Four Polymorphic TDP-43 Amyloid Cores." *Nature Structural & Molecular Biology* 26 (7): 619–27.

3) Figure 1c – The flexible representation of HEWL and Human appear to be identical images. This is difficult to understand, especially with the variation outlined in 1d. Also, the scale appears to be different between flexible and rigid with no reason stated.

4) Page 4 Line 9 – "This finding created an apparent conundrum in which a single protein sequence, which is typically viewed as having a single lowest-energy state"

I believe it is more typically understood that this is not the case and that what constitutes a lower-energy state is environment-dependent.

5) Figure 1d – Why are the mean square deviations over twice as high for the HEWL? They don't have such dissimilar persistence lengths as seen in 1e and this is not explained.

6) Figure S5 – It is mentioned that what appears to be a reversible fiber is extending from an irreversible fiber. This is difficult to reconcile. The width difference between these fibers seems significant. If the model follows from Figure 5, and one assumes the beta-sheet spacing is similar in both reversible and irreversible fibrils, then the mass-per-length would be the same for both; is it assumed that the helical regions do not contribute to the density seen in the AFM? It would be helpful for the authors comment on this.

The interface between the reversible and irreversible is also difficult to visualize and somehow appears counterintuitive to the concept of fibril nucleation. What would the interface of these two structures look like and how dissimilar could they be while maintaining an extended fiber? Does this imply that a transition could occur within the central region of the fiber with no monomeric dissociation and reassociation? It is also difficult to reconcile the purported heterogeneity and dynamics of the reversible fibrils with their regularity and persistence lengths.

7) Page 11 Line 2 – it is unclear how much consistency can be expected between SHIFTX data and measurements. The degree of inaccuracy would compound with resolution accuracy of the cryo-EM model. Perhaps the authors could comment on the drawback of the simulated spectra and their comparison with the reversible fibril spectra.

“A superposition with a simulated 2D spectrum based on the structure of the irreversible fibrils and using typical line widths for fibrils highlights the extent of line broadening of the measured spectra (Fig. 4a).”

The simulated structure accounts for “residues 22-34, 50-128 and an additional 12 residues whose sequence could not be identified, covering more than 2/3 of the amino acid sequence of human lysozyme.” *60% without the unknown 12* How was the “low level” of superposition determined. The degree of superposition appears >50% by inspection. Did the authors try a J-based experiment, such as a TOBSY-INEPT, to probe dynamic regions?

8) Page 4 Line 4 – “This view was, however, challenged by a recent report from our group²⁴ in which human and hen egg-white lysozyme (HEWL), two proteins with only 76% sequence identity, were both found to be capable of folding into distinct amyloids, one flexible and reversible and the other rigid and irreversible.”

It isn't clear whether this is intended to mean human and hen are distinct in forming reversible and irreversible amyloids or that both human and hen can each individually form both reversible and irreversible amyloids. This is also perhaps misleading in the use of “only 76%” in that this is a relatively large percentage and the remaining 24% could be inconsequential to fibril formation.

9) The authors provide CD spectra in support of the reversibility of each fibril. The CD measurements were not described as far as I could tell. A simple secondary structure analysis, such as Dichroweb, would be helpful in characterizing the degree of secondary structure content and supporting the claims (i.e. page 13 line 4).

10) Page 7 Line 12 – “Using standard single particle helical reconstruction techniques we were able to produce an initial de novo 3D model from the 2D classes...”

Single particle and helical reconstruction are opposites. The term “single particle-like” has been applied to a type of reconstruction and may be what the authors are suggesting.

11) Page 10 Line 22 – “Cryo-EM image analysis is consistent with this interpretation, but also indicated that high-resolution structural information could not be obtained with this technique.”

I would recommend the inclusion of some sort of cryo-EM data which indicates this in the supporting material. Was it the case that the authors could not achieve good sample grids or were they able to acquire data, and if so, was this judgment based on inspection of micrographs or a full-fledged attempt to resolve 2D class averages?

12) Page 11 Line 23 – “Usually, a superposition between the PAIN spectrum (blue) and an NC-CP spectrum (purple) shown in Fig. 4b can be used to distinguish between in-register or out-of-register inter-molecular B-sheets..”

A reference describing this method of interpretation would be helpful to include.

13) The authors discuss intermolecular contacts involving alanine and lack of methionine and isoleucine resonances pertaining to SSNMR measurements. I recommend including the primary sequence and highlighting these areas of interest. I would also recommend a discussion on the strict locations of the limited Met content and the Ala content of the sub-domain B with respect to the PAIN experiments.

14) Page 3 Line 21 – The authors refer to reversible amyloid fibrils which are characterized as having low-complexity, yet cryo-EM structures of many of these examples have been solved, e.g.,

Lee, Myungwoon, Ujjayini Ghosh, Kent R. Thurber, Masato Kato, and Robert Tycko. 2020. "Molecular Structure and Interactions within Amyloid-like Fibrils Formed by a Low-Complexity Protein Sequence from FUS." *Nature Communications* 11 (1): 5735.

Lu, Jiahui, Qin Cao, Michael P. Hughes, Michael R. Sawaya, David R. Boyer, Duilio Cascio, and David S. Eisenberg. 2020. "CryoEM Structure of the Low-Complexity Domain of hnRNPA2 and Its Conversion to Pathogenic Amyloid." *Nature Communications* 11 (1): 4090.

Can the authors comment as to why the reversible lysozyme fibrils would behave differently?

Reviewer #2 (Remarks to the Author):

Amyloids are known to form cross-beta fibrils, which, upon reaching a critical energy minimum of self-assembly, commit to forming mature fibers in a potentially irreversible manner due to their very low energy state compared to globular proteins. There may exist a "sweet spot" during self-assembly where fibrils, variably referred to as protofilaments among other names in early stages, are observed but have not yet reached a low enough energy state to prevent disassembly. This stage is likely highly polymorphic, encompassing soluble and insoluble monomers, oligomers, and aggregates in various forms and secondary structures. The study by Frey et al. aims to capture the structures before and after this transition to irreversible fibrils using amyloid models of lysozyme, including both human and hen egg-white orthologs.

Irreversibility was assessed through thermostability, with no change in secondary structure upon heating as observed, supported by AFM measurements indicating a lower level of structural organization. CryoEM determined the fibril structures, revealing distinct arrangements of beta-sheets between human and hen lysozymes, despite a 76% identity between the orthologs. Flexibility within parts of the protein chain was noted. Due to the flexibility and polymorphism, soluble or very early fibril structures could not be determined at high resolution, leading to the use of ssNMR to study this aggregation stage. The ssNMR spectra, when compared to the calculated one from the cryoEM structure and chimera models, showed that the reversible fibrils of human lysozyme have a distinct structure from the irreversible fibrils, and further indicating early stages

composed of both helical and inter-molecular beta-sheet secondary structures. Local helical structure formation may dictate a kinetic trap, favoring reversible over irreversible fibril formation, though reversible fibrils can still become irreversible.

This insight into the fibrillation process and the transient structures on and off the pathway to mature fibrils is invaluable, pursued for years through emerging structural technologies. Considerations include the different protocols and conditions for expressing and purifying human lysozyme for ssNMR and CryoEM studies, the impact of sample preparation on amyloid fibril polymorphism, and the need for clearer methodological descriptions and data presentation in the manuscript. The observed polymorphism in cryo micrographs and the potential for natural variability within the same sample versus differences between early and late-stage fibrils are critical for understanding amyloid fibrillation.

Points of Consideration:

1. The human lysozyme, used to differentiate between reversible and irreversible fibrils via ssNMR and CryoEM, was sourced and processed differently: one commercially for EM and AFM analysis of irreversible fibrils, and the other through yeast expression for NMR and AFM of reversible filaments, with each stored in distinct buffers. Given the significant dependency of amyloid fibril polymorphism on conditions and expression protocols, these preparation differences could influence observed disparities. It raises the question: was the yeast-expressed lysozyme also used for EM experiments?

2. The HEWL exhibited a structure distinct from its human counterpart, potentially due to polymorphism. It prompts inquiry into whether HEWL was also analyzed by ssNMR to assess the distribution of states.

3. The EM methods and supplementary information present challenges in clarity and completeness:

a. Clarification is needed on the pixel size used for data collection, with inconsistencies between the stated and supplementary table values. Specifically, the methods mention that the images were binned by 4 to get a pixel size of 2.6Å but the supplementary table indicates a pixel size of 1.3Å. It seems that this is the pixel size in the postprocessed maps, but was collection done at 1.3Å or 0.65Å? Please clarify that point in the methods. The manuscript should specify if particle images were upsampled from 2.6 to 1.3Å.

b. Corrections are required where RELION is mentioned for motion correction (Page 18), which it does not perform; the use of MotionCor2 (or another program used) should be specified and cited. Also, a dot is missing between Scheres and Relion.

c. The model building section needs clarity on whether it describes initial model creation or amino acid fitting into the map.

d. The absence of 3D classification from the methods contradicts supplementary Table 4.

e. The manuscript lacks FSC curves for half-maps and map versus model comparisons.

f. Resolution estimation methods need detailing, including the chosen FSC value for determination (0.5 or 0.143?).

- g. Table S4 should separate the numbers of fragments post 2D and 3D classification.
 - h. Comprehensive model validation details are missing in Table S4; inclusion of model resolution, composition, and validation metrics is recommended. Please also cite the program used to validate the models.
 - i. The data availability section should include PDB and EM map codes.
4. The extent of polymorphism observed in cryo micrographs and its impact on distinguishing between natural variability and stage-specific differences in fibrils requires elucidation.
 5. The handedness of fibrils as confirmed by AFM analysis is obscured by image binning in Figure 1a; inclusion of higher magnification images could clarify, similar to the one for the human lysozyme in fig S5. Can the author clarify the handedness which seems to be inconsistent between fig S1b (a right-handed twist for the human lysozyme?) and figure 1 and S5 (left-handed fibril?).
 6. Does the reversible fibril show amyloid ~4.8Å and 10-12Å peaks in a diffraction pattern?
 7. The manuscript should define the role indicated by the star symbol in the authors' list.
 8. The term "free-energy" should be presented without a hyphen for consistency (page 3).
 9. Page 9 – “[which is buried because there is another undefined” - is should be are?
 10. The manuscript should clarify whether "beta-sheets" should be referred to as "beta-strands" for accuracy (Pages 7-9).
 11. Discrepancies between described and depicted beta-strands in PDB models for both human lysozyme and HEWL need reconciliation:
Human - b1 stops at 27, b2 starts at 54, b3+b4+b5 are not visible, b6 starts at 83, b7 is missing, b8 starts at 96, b9 starts at 107, b10 starts at 115, b11 stops at 125.
HEWL - strand 39-41 is not in the model, 43-48 stops at 45, 50-52 is missing, 72-78 is missing, 88-90 is 89-91 in the model, 93-100 stops at 99 in model and depending on the repeat, starts at 95.
 12. Figures 2 and 3 lack side views and cross-section projections necessary for validating amyloid reconstruction.
 13. Broad cross peaks (Pages 10-11) suggesting polymorphism in spectral analysis indicate a need for discussion on the assembly's composition.
 14. Page 11: Clarification of the abbreviation NCA is required.
 15. Page 13: A correction in figure referencing aligns CD investigation findings with the correct supplementary figure (S2).
 16. Consistency in abbreviations, such as milliQ versus MQ water and volume expression, should be maintained. wt% - what does it stands for? (space should also be probably added)
 17. When centrifuge speeds are given, conversion to g-force or model specification would aid reproducibility.
 18. Page 17: The number of independent AFM experiments contributing to the analysis of 300 fibrils/filaments should be specified.

19. Page 17: The EM grid preparation section - first sentence – “respectively” is not fitting unless another condition is indicated.

Reviewer #3 (Remarks to the Author):

This manuscript by Frey and Zhou et al. reports a structural study of amyloid filaments formed from lysozyme. The authors prepared two types of filaments formed from both human and hen lysozyme. They then applied AFM, cryo-EM and solid-state NMR to analyse the structural properties of the filaments in the four samples they made in order to study assembly pathways of the lysozyme amyloid structures. It is my opinion that the authors present important fundamental insights on the polymorphic assembly of amyloid structures that they gained from this study through careful and detailed structural characterisation and analysis. The authors present a large quantity and excellent quality structural data from the three complementary structural analysis methods they used. I am particularly excited about the fact that the authors successfully combined AFM, cryo-EM and solid-state NMR together in their detailed analysis to extract meaningful information regarding lysozyme amyloid assembly. I am also thrilled to see the structural comparison between human and hen lysozyme showing different amyloid core folds can be assembled from similar sequences of structurally homologous globular proteins. Overall, therefore, I strongly recommend this manuscript for publication after revisions. Below are my comments, which I hope the authors will find useful in improving their manuscript.

– As mentioned, I am very excited about the combined use of AFM, cryo-EM and solid-state NMR in structural analysis, which in this case offered useful complementary information that together informed about the assembly of lysozyme fibrils. This important aspect is not well discussed and I suggest the authors highlight their combined approach in the general discussions of their work.

– The terms ‘Reversible’ and ‘Irreversible’ fibrils used by the authors is not clearly defined. I understand that the term came from their previous publications (e.g. Cao et al. reference 24) where the curve-linear filaments showed evidence of reversibility following temperature jump experiments. However, ‘Reversible’ and ‘Irreversible’ infers full mechanism and stability information that the authors do not have fully. Therefore, I suggest using the ‘flexible’ and ‘rigid’ wording throughout and/or add a clear definition of the filament nomenclature based on their structural appearance (e.g. curve-linear vs long-straight).

– Cryo-EM experiments may have significant sample and particle picking analysis biases compared to AFM. Therefore, the authors should add evidence to show that the density maps resolved by cryo-EM is in fact the same species as the major filament species seen in the AFM images of the rigid fibrils. This can include comparing the cross-over distances, fibril widths, and/or central line height profiles (see Lutter et al 2022 <https://doi.org/10.1016/j.jmb.2022.167466>). It would also be useful for the readers to know the percentage filaments in the cryo-EM images correspond to the filaments in the final high-res maps.

– The AFM data of the samples are of excellent quality. However, the persistence length is calculated here as a collective property of the population. In this case if there are distinguishable multiple fibril polymorphs in the samples then the persistence should be calculated for each of the

sub-populations. Therefore, the height and the periodicity distributions should be plotted as histograms and not only in box-plots in Fig 1b (this can be done in the SI). This will also help the readers assess the relative size of the main fibril type resolved by the cryo-EM data.

– The information in Figure S5 is interesting and I think it is an important piece of the puzzle. I suggest at least a height distribution analysis of the flexible parts in the mature filament samples and a persistence length comparison to add evidence of whether these parts are likely to be the same as the flexible filaments seen in ‘reversible’ samples.

– Page 14, the authors very briefly discussed “pathway-dependent protein folding/amyloid polymorphism”. This phenomenon was originally described by the Radford laboratory (Gosal et al. *J. Mol. Biol.* (2005) 351, 850–864). I think an extended discussions and speculations on the on/off pathway mechanisms based on the similarities to the behaviours seen in Gosal et al. would be very useful for the readers.

– Heating is required to initiate assembly of lysozyme fibrils, the authors should therefore discuss possible hydrolysis based mechanisms in the assembly. Could the extra densities in the cryo-EM maps be peptide fragments from hydrolysis?

– The ‘reversible’ hen lysozyme sample preparation procedure is missing in the methods section. I assume the procedure is identical to that of human labelled lysozyme?

– The schematic in Figure 5 is confusing. Based on the information in the methods section, should not the first barrier be 80°C t=short (or 5 min) and the second barrier 90°C t = long (or 3 hours)? Also this infers an on-path mechanism? I suggest revising this schematic.

Rebuttal letter and response of reviewers' comments

Reviewer #1: This study presents the argument that a single amino acid sequence, under varied degrees of denaturation, can generate distinct amyloid-like fibrils. The data presented suggest that these fibrils differ in overall morphology, secondary structure content and local dynamics. The model protein, lysozyme, used in this article is a globular protein seen ubiquitously as a standard throughout structural molecular biology. In this study lysozyme is used to understand the energetic landscape by which proteins assemble into stable and often irreversible supramolecular complexes.

The authors present structures of human and HEWL lysozyme fibrils derived from cryo-EM measurements. These were prepared from monomers exposed to “long” heat denaturation and due to the fiber’s resistance to subsequent heat denaturation and rigid homogenous core amenable to cryo-EM investigation, they are referred to as “irreversible” fibrils. The authors also provide SSNMR data for “reversible” fibrils which are classified as such due to their lack of rigidity and sensitivity to heat denaturation. By comparing regions present in the native globular state with those seen in both reversible and irreversible fibrils, the authors conclude that partial denaturation maintains helical structure elements incapable of contributing to canonical beta-sheet fibril formation and that complete denaturation is required for extensive beta-sheet packing present in irreversible fibrils. My comments on the manuscript are as follows:

1) Regarding presentation of cryo-EM data, I note/recommend the following: (1) A density map displaying a half pitch length along the fibrillar axis would be helpful to include and would allow the reader to visualize the degree of resolution achieved along the fibrillar axis; (2) at least one sample micrograph should be included in the supporting material to allow the reader to visualize the quality of the raw data; (3) For figures 2 and 3, the EM data presented in (a) and (b) does not include a scale bar making visual analysis impossible. With twist angles of 1.22 and 1.00, expected half-pitches would be 705.2 Å and 860.4 Å, yet (b) in both figures appear to be the same length. (4) the quality of the 2D classes appear suboptimal when considering the the final resolution.

We thank the reviewer for the comment. We have added 4 orthogonal side views of the cryo-EM reconstructions to Figures S4-S5 in the revised supporting information (SI). Sample micrographs for the irreversible human and hen lysozyme fibrils as well as the reversible (curly) human fibrils have been added to Figures S4,S5 and S6. Scale bars for the 2D classifications and the approximate crossover distance based on the initial model calculation have been added to Figures 2ab and 3ab. The latter differs from the twist in the final refined model and at this point in the structure determination is only an estimate. The quality of 2D classifications is not necessarily indicative of the final resolution achievable, particularly when we only used them to obtain the initial model for 3D refinements. For this initial classification, we chose to work with a large pixel size, both in order to speed up the process but more importantly to get better separation of classes. In our experience, using smaller pixel sizes can lead to a collapsing of the different views into many fewer classes. By using the larger pixel size, we did not need to ignore the CTF until the first peak or increase the amplitude contrast, both options which can lead to better 2D class separation but typically yield lower quality initial models with *reliion_helix_inimodel2d*. In short, the goal of 2D classification was not to get the highest resolution, but to get a good enough separation of classes to remove particles from the straight (untwisted) classes, bad picks or other minor polymorphs and to get class averages that were sufficiently good to make an initial 3D model.

2) Cryo-EM data and processing – In table S4 the pixel size is labeled 1.3 Å, but on page 18 line 14, it says “binned 4x to a pixel size of 2.6 Å”. It was also not mentioned if this was acquired in super resolution mode, which may account for this discrepancy. In the same table there appears to be a marked improvement in the human lysozyme and not in the HEWL with no explanation. The presence of excess electron density and unidentifiable regions of seemingly beta-sheet structures seen in the cryo-EM maps, makes it difficult to access the quality of the structures. The conclusions rest on the foundation of regional comparison between reversible and irreversible structures, yet

both the cryo-EM and SSNMR results seem to be somewhat ambiguous. With the presence of additional unknown electron density, how was it determined what regions belong to rigid structured portion? Could this have been verified with SSNMR? I could not access the EMDB or PDB entries to further assess this, but by inspection, it is difficult to assess how the stated resolution is supported by the data. It would be helpful to include FSC curves in the supporting material. The following provide examples of fibril structures solved to 2.8-3.4 Å resolution:

Wang, Li Qiang, Kun Zhao, Han Ye Yuan, Qiang Wang, Zeyuan Guan, Jing Tao, Xiang Ning Li, et al. 2020. "Cryo-EM Structure of an Amyloid Fibril Formed by Full-Length Human Prion Protein." *Nature Structural & Molecular Biology* 27 (6): 598–602.

Guerrero-Ferreira, Ricardo, Nicholas Mi Taylor, Daniel Mona, Philippe Ringler, Matthias E. Lauer, Roland Riek, Markus Britschgi, and Henning Stahlberg. 2018. "Cryo-EM Structure of Alpha-Synuclein Fibrils." *eLife* 7 (July). <https://doi.org/10.7554/eLife.36402>.

Cao, Qin, David R. Boyer, Michael R. Sawaya, Peng Ge, and David S. Eisenberg. 2019. "Cryo-EM Structures of Four Polymorphic TDP-43 Amyloid Cores." *Nature Structural & Molecular Biology* 26 (7): 619–27.

The methods section for the cryo-EM analysis has been expanded and the ambiguities clarified in the revised manuscript. We have added map-model FSC curves and the side views of the maps in Figures S4-S5 in order to better demonstrate the quality of the models. In regards to the determination of what regions belong to the rigid structured portions of the model, there are two points: first, only the regions that are sufficiently rigid and ordered will give rise to significant density. Thus, the additional unidentified density must also be somewhat rigid and structured for it to appear in the map, albeit to a lesser degree than the sequence-identified portions of the map. The assignment of the sequence to the density was accomplished with ModelAngelo. In the case of the human lysozyme, the map was of sufficient quality that this could be done without supplying the protein sequence to ModelAngelo. The output of ModelAngelo was used as a query in BLASTp to return human lysozyme as the top hit. For the HEWL map, the protein sequence needed to be supplied to ModelAngelo and the output model fit very well to the density. This explanation has been added to the methods section.

3) Figure 1c – The flexible representation of HEWL and Human appear to be identical images. This is difficult to understand, especially with the variation outlined in 1d. Also, the scale appears to be different between flexible and rigid with no reason stated.

We are most thankful to the reviewer for having spotted this mistake. We have erroneously used the same panel for HEWL and human lysozyme. We apologize for this mistake and have now fixed it.

4) Page 4 Line 9 – "This finding created an apparent conundrum in which a single protein sequence, which is typically viewed as having a single lowest-energy state" I believe it is more typically understood that this is not the case and that what constitutes a lower-energy state is environment-dependent.

We agree with the referee that ultimately are the folding conditions, therefore the environment and temperature, which determine the lowest-energy state. That is why we refer to as *apparent conundrum*. In the revised text, we have put the work "*apparent*" in italic.

5) Figure 1d – Why are the mean square deviations over twice as high for the HEWL? They don't have such dissimilar persistence lengths as seen in 1e and this is not explained.

We thank the reviewer for pointing this out. We also noted the mean square deviations in Figure 1d, the HEWL (lower panel) is twice higher over the human lysozyme (upper panel). We believe this is related to the intrinsic polymeric properties of these fibrils, which means that the amyloid core in the human lysozyme irreversible fibrils is more rigid than that of HEWL irreversible fibrils. Furthermore,

the mean square deviations in Figure 1d are obtained by the software EasyWorm according to the fibril shape fluctuation (Figure 1c), which, nonetheless, also indicates human lysozyme slightly more rigid.

In fact, the Figure 1e shows the persistence length acquired by another software (FiberApp) that calculate the mean squared end-to-end distance between contour segments, indicating 3313 ± 128 and 8810 ± 203 nm for HEWL and human lysozyme irreversible fibrils, respectively. These results from two different algorithms confirm the dissimilarity in the rigidity of the amyloid core between HEWL and human lysozyme irreversible fibrils. The different software and different fibrils properties may possibly account for differences in mean square deviations.

6) Figure S5 – It is mentioned that what appears to be a reversible fiber is extending from an irreversible fiber. This is difficult to reconcile. The width difference between these fibers seems significant. If the model follows from Figure 5, and one assumes the beta-sheet spacing is similar in both reversible and irreversible fibrils, then the mass-per-length would be the same for both; is it assumed that the helical regions do not contribute to the density seen in the AFM? It would be helpful for the authors comment on this.

Thanks for the comment. However, we argue that AFM cannot be used to reliably estimate the mass-per-length, due to its intrinsic convolution effect. Furthermore, what Figure S5 (Figure S10 in the revised manuscript) gives is the heights, which can be different for rigid and soft fibrils with identical mass-per-length within the same fibril.

The interface between the reversible and irreversible is also difficult to visualize and somehow appears counterintuitive to the concept of fibril nucleation. What would the interface of these two structures look like and how dissimilar could they be while maintaining an extended fiber? Does this imply that a transition could occur within the central region of the fiber with no monomeric dissociation and reassociation? It is also difficult to reconcile the purported heterogeneity and dynamics of the reversible fibrils with their regularity and persistence lengths.

We acknowledge the reviewer for this comment. The folding motif can change within a few Armstrong along the contour length of the same fibril, while preserving the cross- β periodicity. In the end, the flexible fibrils are loose folding versions of rigid fibrils. It is also possible that the folding of each β -strands changes in-plane orthogonal to the fibril axis while maintaining the same inter β -strands distance without conflict with the concept of nucleation.

7) Page 11 Line 2 – it is unclear how much consistency can be expected between SHIFTX data and measurements. The degree of inaccuracy would compound with resolution accuracy of the cryo-EM model. Perhaps the authors could comment on the drawback of the simulated spectra and their comparison with the reversible fibril spectra. “A superposition with a simulated 2D spectrum based on the structure of the irreversible fibrils and using typical line widths for fibrils highlights the extent of line broadening of the measured spectra (Fig. 4a).”

Thanks for the comment. The accuracy/resolution of the EM reconstructions is not having a large impact on the accuracy of the predictions because the latter are dependent mainly on the secondary structure. Also, the accuracy of the predictions is within linewidth (1 ppm). An explanation has been added to the main text in the revised manuscript (Page 11).

The simulated structure accounts for “residues 22-34, 50-128 and an additional 12 residues whose sequence could not be identified, covering more than 2/3 of the amino acid sequence of human lysozyme.” *60% without the unknown 12* How was the “low level” of superposition determined. The degree of superposition appears >50% by inspection. Did the authors try a J-based experiment, such as a TOBSY-INEPT, to probe dynamic regions?

The indicated superposition by the reviewer likely included the side chain resonances (higher than CB) of solvent exposed residues that usually show structure-independent chemical shifts and are as such not useful for a fingerprint of the structure in contrast to the CA/CB chemical shifts that are sensitive to the secondary structure and hydrophobic side chains that are sensitive to the core structure.

The “low level” of superposition is of qualitative nature and should “only” highlight that the cryo EM fibril structure is not present in the reversible fibrils. This procedure is now described in the main manuscript by “low level of superposition by visual inspection”.

A J-based experiment was not recorded and can not be recorded anymore due to sample age and the problem that the sample production will take several of months of labor for an expected little increase in information since lack of density at the cryo EM level can not be related directly to flexibility.

8) Page 4 Line 4 – “This view was, however, challenged by a recent report from our group²⁴ in which human and hen egg-white lysozyme (HEWL), two proteins with only 76% sequence identity, were both found to be capable of folding into distinct amyloids, one flexible and reversible and the other rigid and irreversible.”

It isn't clear whether this is intended to mean human and hen are distinct in forming reversible and irreversible amyloids or that both human and hen can each individually form both reversible and irreversible amyloids. This is also perhaps misleading in the use of “only 76%” in that this is a relatively large percentage and the remaining 24% could be inconsequential to fibril formation.

We agree that the sentence is ambiguous, we have removed the word “only” in the revised manuscript.

9) The authors provide CD spectra in support of the reversibility of each fibril. The CD measurements were not described as far as I could tell. A simple secondary structure analysis, such as Dichroweb, would be helpful in characterizing the degree of secondary structure content and supporting the claims (i.e. page 13 line 4).

We appreciate the reviewer for the comment. In the revised manuscript, we applied the BestSel analysis (<https://doi.org/10.1073/pnas.1500851112>) to illustrate the β -sheet conformation variation with the heat treatment. As seen in temperature dependent CD measurements of Fig. S3 in the revised SI, the β -sheet content of flexible fibril reduced upon heat treatment, while the β -sheet content of rigid fibril remains constant. More result on the reversibility can be seen on our earlier paper (Cao et al, JACS 2021).

Following the Reviewer's suggestion, we also analyzed the secondary structure conformation of reversible fibril in Figure S3 in the revised Supporting information, demonstrating the mixed conformation of helical and β -sheet conformation, which is supporting the claims in the manuscript text.

10) Page 7 Line 12 – “Using standard single particle helical reconstruction techniques we were able to produce an initial de novo 3D model from the 2D classes...”

Single particle and helical reconstruction are opposites. The term “single particle-like” has been applied to a type of reconstruction and may be what the authors are suggesting.

Thank you for this correction. It has been changed to “Using standard single-particle based helical reconstruction techniques” in the revised manuscript (Page 7).

11) Page 10 Line 22 – “Cryo-EM image analysis is consistent with this interpretation, but also indicated that high-resolution structural information could not be obtained with this technique.”

I would recommend the inclusion of some sort of cryo-EM data which indicates this in the supporting material. Was it the case that the authors could not achieve good sample grids or were they able to acquire data, and if so, was this judgment based on inspection of micrographs or a full-fledged attempt to resolve 2D class averages?

We attempted to resolve 2D class averages but were not able to get a reasonable resolution of classes and the obtained classes did not contain any high-resolution data. This has now been elaborated on in the text and representative micrographs and the 2D classes added to Figure S6 in the revised SI.

12) Page 11 Line 23 – “Usually, a superposition between the PAIN spectrum (blue) and an NC-CP spectrum (purple) shown in Fig. 4b can be used to distinguish between in-register or out-of-register inter-molecular B-sheets..”

A reference describing this method of interpretation would be helpful to include.

We thank the reviewer for this comment. The following reference has been cited in the revised manuscript as requested (Page 11).

Wälti MA, Ravotti F, Arai H, Glabe CG, Wall JS, Böckmann A, Güntert P, Meier BH, Riek R. *Proc Natl Acad Sci USA*. 2016 ;113(34):E4976-84. doi: 10.1073/pnas.1600749113.

13) The authors discuss intermolecular contacts involving alanine and lack of methionine and isoleucine resonances pertaining to SSNMR measurements. I recommend including the primary sequence and highlighting these areas of interest. I would also recommend a discussion on the strict locations of the limited Met content and the Ala content of the sub-domain B with respect to the PAIN experiments.

We thank the reviewer for this comment. We have added a new panel to Figure 4 (g), highlighting the structural elements for both fibril and monomer forms in the primary sequence of lysozyme, as well as add the following text to explain how the Met, Ile, and Ala peaks in the spectra support our model. “... the simulated spectra of the hypothetical chimera structure resemble the experimentally measured NMR spectra reasonably well (Figs. 4f and S8c) for example the position of the Ala cross peaks in the 2D DARR discussed above and indicated in Fig. 4. It is also noted that predicted cross peaks for Met and Ile (highlighted by yellow and red crosses, respectively in Fig. 4e) are absent in the measured spectra while the corresponding diagonals are observed. This loss of expected signals indicates the presence of intermediate dynamics that further hint at the overall dynamics of the helical segments, as both Met and Ile are mostly localized within subdomain A.” in the revised manuscript (Page 14).

14) Page 3 Line 21 – The authors refer to reversible amyloid fibrils which are characterized as having low-complexity, yet cryo-EM structures of many of these examples have been solved, e.g., Lee, Myungwoon, Ujjayini Ghosh, Kent R. Thurber, Masato Kato, and Robert Tycko. 2020. “Molecular Structure and Interactions within Amyloid-like Fibrils Formed by a Low-Complexity Protein Sequence from FUS.” *Nature Communications* 11 (1): 5735.

Lu, Jiahui, Qin Cao, Michael P. Hughes, Michael R. Sawaya, David R. Boyer, Duilio Cascio, and David S. Eisenberg. 2020. “CryoEM Structure of the Low-Complexity Domain of hnRNPA2 and Its Conversion to Pathogenic Amyloid.” *Nature Communications* 11 (1): 4090.

Can the authors comment as to why the reversible lysozyme fibrils would behave differently?

We respectfully disagree with this point. We see no contradiction between our statement and what is reported in the literature above. Furthermore, we note that the molecular polymorphism of reversible amyloid fibril is by definition sequence dependent, and thus any extrapolation/generalization may be speculative.

Reviewer #2 (Remarks to the Author):

Amyloids are known to form cross-beta fibrils, which, upon reaching a critical energy minimum of self-assembly, commit to forming mature fibers in a potentially irreversible manner due to their very low energy state compared to globular proteins. There may exist a "sweet spot" during self-assembly where fibrils, variably referred to as protofilaments among other names in early stages, are observed but have not yet reached a low enough energy state to prevent disassembly. This stage is likely highly polymorphic, encompassing soluble and insoluble monomers, oligomers, and aggregates in various forms and secondary structures. The study by Frey et al. aims to capture the structures before and after this transition to irreversible fibrils using amyloid models of lysozyme, including both human and hen egg-white orthologs.

Irreversibility was assessed through thermostability, with no change in secondary structure upon heating as observed, supported by AFM measurements indicating a lower level of structural organization. CryoEM determined the fibril structures, revealing distinct arrangements of beta-sheets between human and hen lysozymes, despite a 76% identity between the orthologs. Flexibility within parts of the protein chain was noted. Due to the flexibility and polymorphism, soluble or very early fibril structures could not be determined at high resolution, leading to the use of ssNMR to study this aggregation stage. The ssNMR spectra, when compared to the calculated one from the cryoEM structure and chimera models, showed that the reversible fibrils of human lysozyme have a distinct structure from the irreversible fibrils, and further indicating early stages composed of both helical and inter-molecular beta-sheet secondary structures. Local helical structure formation may dictate a kinetic trap, favoring reversible over irreversible fibril formation, though reversible fibrils can still become irreversible.

This insight into the fibrillation process and the transient structures on and off the pathway to mature fibrils is invaluable, pursued for years through emerging structural technologies. Considerations include the different protocols and conditions for expressing and purifying human lysozyme for ssNMR and CryoEM studies, the impact of sample preparation on amyloid fibril polymorphism, and the need for clearer methodological descriptions and data presentation in the manuscript. The observed polymorphism in cryo micrographs and the potential for natural variability within the same sample versus differences between early and late-stage fibrils are critical for understanding amyloid fibrillation.

Points of Consideration:

1. The human lysozyme, used to differentiate between reversible and irreversible fibrils via ssNMR and CryoEM, was sourced and processed differently: one commercially for EM and AFM analysis of irreversible fibrils, and the other through yeast expression for NMR and AFM of reversible filaments, with each stored in distinct buffers. Given the significant dependency of amyloid fibril polymorphism on conditions and expression protocols, these preparation differences could influence observed disparities. It raises the question: was the yeast-expressed lysozyme also used for EM experiments?

The atomic polymorphism of commercial and expressed human lysozyme is never compared directly: the commercial human lysozyme was used for *rigid fibrils* solved by cryoEM while the expressed lysozyme was used to provide insight on the structure of *flexible fibrils* by ssNMR. That the structure among these two forms of the fibrils must be totally different is evident by the mesoscopic polymorphism itself. In this respect, we did control by AFM the mesoscopic polymorphism of both commercial and expressed human lysozyme irreversible fibrils, and we found the two behaviors to be identical. Thus, the expression of human lysozyme in yeast was necessary only to provide single/double-labeled protein, for ssNMR studies. This has no implication on the atomic resolution of the structure achieved by cryoEM. Furthermore, we note that the preparation protocol and buffer for the preparation of the amyloid of human lysozyme used for the cryoEM (commercial) and ssNMR

(expressed) studies were identical, therefore, we have no reasons to believe the structural information is isotope-dependent. We would like to point out that the successful expression and purification of single and double labeled human lysozyme in 33.6 mg batch quantity (the minimum quantity needed for our ssNMR rotor) required several attempts over a length of two years and that the remaining double labelled lysozyme after ssNMR studies was not enough to produce rigid fibrils for cryoEM using the same protocol that we had previously used. We hope the reviewer will appreciate the significant work involved.

2. The HEWL exhibited a structure distinct from its human counterpart, potentially due to polymorphism. It prompts inquiry into whether HEWL was also analyzed by ssNMR to assess the distribution of states.

As explained in the point above, two years were invested (and several tens of thousands of euros for isotope labelling) for the expression of single/double labeled human lysozyme only. We had neither resources nor time available to attempt the same effort for HEWL.

3. The EM methods and supplementary information present challenges in clarity and completeness:
- Clarification is needed on the pixel size used for data collection, with inconsistencies between the stated and supplementary table values. Specifically, the methods mention that the images were binned by 4 to get a pixel size of 2.6Å but the supplementary table indicates a pixel size of 1.3Å. It seems that this is the pixel size in the postprocessed maps, but was collection done at 1.3Å or 0.65Å? Please clarify that point in the methods. The manuscript should specify if particle images were upscaled from 2.6 to 1.3Å.
 - Corrections are required where RELION is mentioned for motion correction (Page 18), which it does not perform; the use of MotionCor2 (or another program used) should be specified and cited. Also, a dot is missing between Scheres and Relion.
 - The model building section needs clarity on whether it describes initial model creation or amino acid fitting into the map.
 - The absence of 3D classification from the methods contradicts supplementary Table 4.
 - The manuscript lacks FSC curves for half-maps and map versus model comparisons.
 - Resolution estimation methods need detailing, including the chosen FSC value for determination (0.5 or 0.143?).
 - Table S4 should separate the numbers of fragments post 2D and 3D classification.
 - Comprehensive model validation details are missing in Table S4; inclusion of model resolution, composition, and validation metrics is recommended. Please also cite the program used to validate the models.
 - The data availability section should include PDB and EM map codes.

We sincerely thank the reviewer for the comment on the EM methods. We have made the corrections on these points with a point-by-point response as following:

- We have expanded the methods section to clarify these ambiguities. As the reviewer suspected, the images were collected at 0.65 Å/pixel and the particles downsampled to 2.6 for 2D classification and then to 1.3 Å for refinement.
- We used Relion's own implementation of the MotionCorr2 package and therefore did not think it was necessary to reference separately. A reference has been added.
- We have expanded this section to clarify the steps taken in the model building process.
- The methods section now contains a more complete explanation of the particle curation including 3D classification.
- e/f. These have been added to Figure S4-5 and the FSC cutoff used (0.143) has been included.
- g./h. The requested changes have been made to Table S4.
- The PDB and EMD codes have been added to the data availability section.

4. The extent of polymorphism observed in cryo micrographs and its impact on distinguishing between natural variability and stage-specific differences in fibrils requires elucidation.

There are other fibril polymorphs in the micrographs whose presence become apparent after 2D classification. These are mostly non-twisted filaments that cannot be used for helical reconstruction. However, in the case of the human lysozyme irreversible fibrils, we looked more into this and were able to solve the structure of a contaminant protein that happened formed fibrils in the same sample. By using the filament subset selection tool in Relion 5, which became available after our initial submission, we were able to separate 2D classes for and solve the structure of this minor contaminant that was present at less than 10% in the batch of commercial lysozyme (Sigma product # L1667) that we used for this study. At first we thought that the fibrils could be another lysozyme polymorph but the lysozyme sequence did not fit into the density. We therefore analyzed the sample by mass spectrometry and identified a rice protein (trypsin inhibitor) whose sequence did fit well into the density. Our previous work on the human lysozyme (JACS 2021) was from a different batch of protein that did not have detectable levels of this contaminant protein by MS and so we conclude that its presence does not impact the results of this study. We plan to publish the structure of the contaminant protein separately as it does not pertain to the results of this study.

5. The handedness of fibrils as confirmed by AFM analysis is obscured by image binning in Figure 1a; inclusion of higher magnification images could clarify, similar to the one for the human lysozyme in fig S5. Can the author clarify the handedness which seems to be inconsistent between fig S1b (a right-handed twist for the human lysozyme?) and figure 1 and S5 (left-handed fibril?).

We are most thankful to the reviewer for raising this point. The rigid fibrils of both HEWL and human lysozyme are left-handed, while the flexible fibrils showed no clear periodicity along the fibril contour length, and thus no resolvable handedness. We have updated AFM images (Figure 1a) with higher magnification in the revised manuscript. The inconsistency found in Fig. S1b was due to the mistakenly flipping of image while arranging the scale bar in the figure. In the revised supporting information, the higher-resolution AFM image has been updated in Figure S1b, in which it demonstrates the left-handed rigid fibrils. We are sincerely thankful to the Reviewer for his/her careful analysis of the figure and for having spotted this error.

6. Does the reversible fibril show amyloid ~4.8Å and 10-12Å peaks in a diffraction pattern?

Yes, this is a solved problem. This can be found in Fig. 3d for human lysozyme and Fig. S13 (for HEWL lysozyme) of our paper in JACS 2021. In short, both reversible and rigid fibril show those ~4.8Å and 10-12Å peaks in X-ray diffraction.

7. The manuscript should define the role indicated by the star symbol in the authors' list.

Thanks for spotting this typo. This has now been fixed in the revised manuscript. The star referred to corresponding authorship.

8. The term "free-energy" should be presented without a hyphen for consistency (page 3).

Thanks for noticing this. The “free energy” has now been adapted in the revised manuscript.

9. Page 9 – “[which is buried because there is another undefined” - is should be are?

This refers only to the last Asn in the list (Asn77).

10. The manuscript should clarify whether "beta-sheets" should be referred to as "beta-strands" for accuracy (Pages 7-9).

We have changed two instances of beta-sheet to beta-strand in the revised manuscript (Pages 7-9). Otherwise, our use of beta-sheet is the correct term. Thank you for spotting this!

11. Discrepancies between described and depicted beta-strands in PDB models for both human lysozyme and HEWL need reconciliation:

Human - b1 stops at 27, b2 starts at 54, b3+b4+b5 are not visible, b6 starts at 83, b7 is missing, b8 starts at 96, b9 starts at 107, b10 starts at 115, b11 stops at 125.

HEWL - strand 39-41 is not in the model, 43-48 stops at 45, 50-52 is missing, 72-78 is missing, 88-90 is 89-91 in the model, 93-100 stops at 99 in model and depending on the repeat, starts at 95.

Despite the widespread use of the DSSP algorithm for secondary structure assignments, the definition of secondary structural elements is not an exact science. Also, while it was often clear during the model-building process which regions of the fibril structures contained beta-strands, we did not apply any secondary structure restraints in ISOLDE or Phenix during the structure refinements except for the regions that had been identified by ISOLDE during the refinement. Therefore, some backbone hydrogen bonds are not idealized beta conformations despite the fact these hydrogen bonds certainly exist (energetically too costly for them to be unfulfilled). Also, the symmetry restraints applied during ISOLDE were released during the Phenix refinement and so subtle differences can exist within the different layers of the model. The ambiguity with secondary structure definition is also apparent by comparing the output of various algorithms like STRIDE (Proteins: Structure, Function, and Genetics 23:566-579 (1995)), the Amyloid Illustrator (used in Figures 2 and 3) and the graphical output during the actual PDB deposition, all of which display small differences from each other and with DSSP. To limit the ambiguity, we have stated in the revised manuscript (Page 9) that we use the STRIDE definitions for the central strand of the deposited coordinates. For human lysozyme these are: β 1 23-32, β 2 51-58, β 3 62-63, β 4 68-69, β 5 74-75, β 6 78-79, β 7 83-87, β 8 96-101, β 9 107-112, β 10 115-119, β 11 120-125 and for HEWL: β 1 28-30, β 2 33-38, β 3 43-45, β 4 51-52, β 5 55-62, β 6 73-76, β 7 80-83, β 8 89-92, β 9 93-98.

12. Figures 2 and 3 lack side views and cross-section projections necessary for validating amyloid reconstruction.

We thank the reviewer for the comment and have added four side views of the cryo-EM maps in Figure S4-5 in the revised manuscript and SI.

13. Broad cross peaks (Pages 10-11) suggesting polymorphism in spectral analysis indicate a need for discussion on the assembly's composition.

The interpretation of the broad cross peaks based on the relaxation data indicate the presence of structural heterogeneity and dynamics as discussed in the text under paragraph "Reversible fibrils of human lysozyme are heterogeneous and dynamic" and "Constructing a plausible 3D model of the reversible fibrils of human lysozyme". In the revised version of the manuscript (Page 11) it is now furthermore stated that "... the presence of several polymorphs in the sample cannot account for the broad peaks because they usually yield multiple distinct cross peaks per ^{13}C - ^{13}C moiety." In extremis, the structural heterogeneity can be interpreted as a very large amount of polymorphs, but we think this interpretation is not meaningful and we would like rather to stick with structural heterogeneity.

14. Page 11: Clarification of the abbreviation NCA is required.

Following the suggestion by the reviewer, "NCA" is replaced by the following phrase: "in the region that correlates nitrogen with alpha carbon resonances (Figure 4, indicated area between 48-60 ppm)" (Page 11).

15. Page 13: A correction in figure referencing aligns CD investigation findings with the correct supplementary figure (S2).

We thank the reviewer for spotting this typo, and it's fixed in the revised manuscript.

16. Consistency in abbreviations, such as milliQ versus MQ water and volume expression, should be maintained. wt% - what does it stand for? (space should also be probably added)

We sincerely thank the reviewer for checking these points. The abbreviations and expressions have been updated to ensure consistency throughout the revised manuscript. And wt% refers to the unit of weight percent.

17. When centrifuge speeds are given, conversion to g-force or model specification would aid reproducibility.

We confirm that the unit of centrifugation has converted into g-force in the revised manuscript.

18. Page 17: The number of independent AFM experiments contributing to the analysis of 300 fibrils/filaments should be specified.

We confirm that the specifications of AFM images used for the statistical analysis has been updated in the revised manuscript.

19. Page 17: The EM grid preparation section - first sentence – “respectively” is not fitting unless another condition is indicated.

Thank you for noticing this mistake. It has been corrected.

Reviewer #3 (Remarks to the Author):

This manuscript by Frey and Zhou et al. reports a structural study of amyloid filaments formed from lysozyme. The authors prepared two types of filaments formed from both human and hen lysozyme. They then applied AFM, cryo-EM and solid-state NMR to analyse the structural properties of the filaments in the four samples they made in order to study assembly pathways of the lysozyme amyloid structures. It is my opinion that the authors present important fundamental insights on the polymorphic assembly of amyloid structures that they gained from this study through careful and detailed structural characterisation and analysis. The authors present a large quantity and excellent quality structural data from the three complementary structural analysis methods they used. I am particularly excited about the fact that the authors successfully combined AFM, cryo-EM and solid-state NMR together in their detailed analysis to extract meaningful information regarding lysozyme amyloid assembly. I am also thrilled to see the structural comparison between human and hen lysozyme showing different amyloid core folds can be assembled from similar sequences of structurally homologous globular proteins. Overall, therefore, I strongly recommend this manuscript for publication after revisions. Below are my comments, which I hope the authors will find useful in improving their manuscript.

– As mentioned, I am very excited about the combined use of AFM, cryo-EM and solid-state NMR in structural analysis, which in this case offered useful complementary information that together informed about the assembly of lysozyme fibrils. This important aspect is not well discussed and I suggest the authors highlight their combined approach in the general discussions of their work.

We thank the reviewer for his/her appreciation of this work. We agree to highlight this and have modified the text in the abstract and the introduction sections in the revised manuscript.

– The terms ‘Reversible’ and ‘Irreversible’ fibrils used by the authors is not clearly defined. I understand that the term came from their previous publications (e.g. Cao et al. reference 24) where the curve-linear filaments showed evidence of reversibility following temperature jump experiments. However, ‘Reversible’ and ‘Irreversible’ infers full mechanism and stability information that the authors do not have fully. Therefore, I suggest using the ‘flexible’ and ‘rigid’ wording throughout and/or add a clear definition of the filament nomenclature based on their structural appearance (e.g. curve-linear vs long-straight).

We agree with this comment. However, flexible and rigid are vague terms to be used in this context vs reversible/irreversible. What we have done in the revised manuscript is the following: when introducing our reversible and irreversible lysozyme fibril for the first time in the text (Page 4), we have made sure to specify that we refer to thermal labile behavior.

– Cryo-EM experiments may have significant sample and particle picking analysis biases compared to AFM. Therefore, the authors should add evidence to show that the density maps resolved by cryo-EM is in fact the same species as the major filament species seen in the AFM images of the rigid fibrils. This can include comparing the cross-over distances, fibril widths, and/or central line height profiles (see Lutter et al 2022 <https://doi.org/10.1016/j.jmb.2022.167466>). It would also be useful for the readers to know the percentage filaments in the cryo-EM images correspond to the filaments in the final high-res maps.

Although there can be significant effects due to the substate in AFM, we find that the helical twist measured by AFM for both the human and hen lysozyme irreversible fibrils is very often similar to that as determined by Cryo-EM. During the manual filament selection in the micrographs, we did not attempt to pick any particular polymorph and so the total number of picked particles compared to the number of particles used in the final reconstruction is somewhat indicative of what percentage of filaments in the cryo-EM images correspond to the refined models. In fact, this is a lower limit because the particle classification steps remove some particles from nearly every picked filament. It is therefore difficult to put a precise number on the percent of filaments in the micrographs that contribute to the final map (or, in other words, any percentage number would be rather inaccurate).

– The AFM data of the samples are of excellent quality. However, the persistence length is calculated here as a collective property of the population. In this case if there are distinguishable multiple fibril polymorphs in the samples then the persistence should be calculated for each of the sub-populations. Therefore, the height and the periodicity distributions should be plotted as histograms and not only in box-plots in Fig 1b (this can be done in the SI). This will also help the readers assess the relative size of the main fibril type resolved by the cryo-EM data.

We appreciate the comment from the reviewer and we generally do agree of course. Indeed, when there is height variance related to different number of protofilaments, that is in presence of mesoscopic polymorphism, fibrils need to be grouped by each group of protofilament number, corresponding to a distinct class of morphology for persistence length calculation. We actually were among those having pioneered this method for AFM (Adamcik et al Nature Nanotechnology 2010). In the present case, however, all fibrils are formed by a homogenous single twisting protofilament, therefore this subgrouping is not possible. The spread in height distribution in the present case has only statistical nature, i.e. does not correspond to different morphologies for which such subgrouping would be necessary. Consequently, also persistence length is an average from a single (spread) population.

– The information in Figure S5 is interesting and I think it is an important piece of the puzzle. I suggest at least a height distribution analysis of the flexible parts in the mature filament samples and

a persistence length comparison to add evidence of whether these parts are likely to be the same as the flexible filaments seen in 'reversible' samples.

We thank the reviewer for this comment. Following the reviewer's suggestion, we selectively investigated the flexible part and rigid part on the exact same fibrils, as seen in Figure S5 (current Figure S10 in the revised SI). We found that both flexible part and rigid part on these fibrils maintain the characteristic features, i.e. shape, height, persistence length, of pure flexible fibrils and pure rigid fibrils. Notably, the flexible part in the fibrils showed a low persistence length (106 nm), which is nearly two orders of magnitude lower than the rigid fibril (8060 nm). This evidence reveals that the flexible part in the fibril is of the same nature of the pure flexible reversible fibrils.

– Page 14, the authors very briefly discussed “pathway-dependent protein folding/amyloid polymorphism”. This phenomenon was originally described by the Radford laboratory (Gosal et al. J. Mol. Biol. (2005) 351, 850–864). I think an extended discussions and speculations on the on/off pathway mechanisms based on the similarities to the behaviours seen in Gosal et al. would be very useful for the readers.

We agree with the reviewer. Following the suggestion, a further discussion on this mechanism has been added in the discussion section in the revised manuscript (Page 14).

– Heating is required to initiate assembly of lysozyme fibrils, the authors should therefore discuss possible hydrolysis based mechanisms in the assembly. Could the extra densities in the cryo-EM maps be peptide fragments from hydrolysis?

We have confirmed that the extra densities have no connection with the protein hydrolysis. We have performed the MALDI-TOF mass spectroscopy (as seen in Figure S9 in the revised supporting information, in line with the result in reference Cao et al, JACS 2020) and confirmed no hydrolysis under this incubation and fibrillization condition.

– The 'reversible' hen lysozyme sample preparation procedure is missing in the methods section. I assume the procedure is identical to that of human labelled lysozyme?

Thank you for noting this. The HEWL lysozyme flexible fibril preparation has been updated in the method section.

– The schematic in Figure 5 is confusing. Based on the information in the methods section, should not the first barrier be 80°C t=short (or 5 min) and the second barrier 90°C t = long (or 3 hours)? Also this infers an on-path mechanism? I suggest revising this schematic.

We appreciate the reviewer for noticing this mistake. This mistake has been corrected in the schematic in Figure 5 in the revised manuscript.

REVIEWER COMMENTS

Reviewer #2 (Remarks to the Author):

Comment 1: Regarding the preparation and conditions used for the two methods, we appreciate the substantial effort invested in both methods and the structure determination. However, the primary argument of the paper is to compare and differentiate between reversible and irreversible fibrils, whereas the structure was obtained using two different methods. One cannot help but wonder if the observed differences arise from the preparation and conditions, which is often the case with amyloids.

The authors responded, stating, "Furthermore, we note that the preparation protocol and buffer for the preparation of the amyloid of human lysozyme used for the cryoEM (commercial) and ssNMR (expressed) studies were identical." However, upon comparing the buffer composition between the two versions, we noticed a discrepancy in the incubation temperature for the rigid fibrils of human lysozyme (see Table attached).

Additionally, the literature (<https://doi.org/10.1016/j.bpc.2023.106962>) suggests that lysozyme fibrillation depends on DTT concentration. We observed that the DTT concentration changes between the rigid and flexible states, with no clear DTT concentration specified for the flexible fibrils.

The authors also stated, "The atomic polymorphism of commercial and expressed human lysozyme is never compared directly." However, the subsequent conclusions are indeed compared. The authors assume that only the incubation time matters, suggesting in the final model that reversible morphologies form first, followed by irreversible fibrils at a later stage. Given the different buffers due to varying DTT concentrations, which can influence lysozyme's ability to form amyloid fibrils, time cannot be considered the sole parameter. Therefore, the assumption that reversible fibrils form first, followed by rigid ones, should be stated more cautiously, clarifying that the proposed mechanism is still under examination.

Comment 3: Thank you for clarifying the cryo-EM methods. It has become much clearer. However, there appears to be confusion between the generation of the first 3D volume (initial model) and model building, which involves generating the PDB file with the 3D alignment of the amino acid sequence.

The image processing section must include all data processing steps from motion correction to postprocessing, including the initial model, 3D classification, 3D refinement, CTF refinement, Bayesian polishing, and postprocessing. Therefore, the section from "Alignments of the segments from the selected 2D classes" to "such mask were used in a final postprocessing step" should be moved to the image processing section. Additionally, please correct the typo "mask were" to "mask was."

Comment 4: It is crucial to at least mention this impurity, which might affect the results concerning irreversible fibril formation. Even if the impurity was not detected in the previous study, it does not

mean it was not present at very low concentrations. Furthermore, the minimum concentration required to trigger lysozyme irreversible fibril formation is unknown.

BTW, if a comment is unclear, please contact the editor for clarification instead of making assumptions, as this ensures an accurate response. Regarding the polymorphism observed in the sample by cryo-EM, the key question is whether this polymorphism is due to incubation time, as suggested, or due to natural variability such as buffer composition or temperature. If the exact same conditions are used (which is not the case here), it becomes feasible to compare the reversible and irreversible states in the model suggested in Figure 5.

Comment 18: The question was about the number of independent experiments. In the revised version, where you indicated the number of areas on the mica, you have a typo :” fibri”.

Few more typos detected:

- Abstract: combing should be combining
- Fig S4 and S5: “the a map”.
- Methods: “For the expressed human lysozyme reversible fibrils used in solid-state NMR, the protein were purified.”

Human lysozyme	Version 1	Version 2
Commercial	Rigid fibrils: 20 mg/mL 100 mM DTT 10 mM NaCl pH 7 80C for 3 h, 300 rpm Ice bath for 30 min Stored at 4C Flexible fibrils: ????	Rigid fibrils: 20 mg/mL 100 mM DTT 10 mM NaCl pH 7 85C for 3 h, 300 rpm Ice bath for 30 min Stored at 4C Flexible fibrils: 20 mg/mL 20-100 mM DTT 10 mM NaCl pH 7 80-85C for 5 min, 300 rpm Ice bath for 30 min Stored at 4C
ssNMR	20 mM DTT 10 mM NaCl pH 7 80C for 5 min, 300 rpm RT to cool down Stored at 4C	20 mM DTT 10 mM NaCl pH 7 80C for 5 min, 300 rpm RT to cool down Stored at 4C
HEWL	Version 1	Version 2
Commercial	Rigid fibrils: 20 mg/mL 100 mM DTT 10 mM NaCl pH 7 90C for 3 h, 300 rpm Ice bath for 30 min Stored at 4C Flexible fibrils: ????	Rigid fibrils: 20 mg/mL 100 mM DTT 10 mM NaCl pH 7 90C for 5 min, 300 rpm Ice bath for 30 min Stored at 4C Flexible fibrils: 20 mg/mL 20-100 mM DTT 10 mM NaCl pH 7 85-90C for 5 min, 300 rpm Ice bath for 30 min Stored at 4C
ssNMR		

Reviewer #3 (Remarks to the Author):

The authors have addresses all of my comments and I think this revised manuscript is improved and ready for publication.

Reviewer #5 (Remarks to the Author):

The concerns of reviewer #1 in regard to NMR experiments have been thoroughly addressed. In terms of the technical description of the measurements and discussion, the manuscript is ready for publication.

In particular,

- point 7 has been clarified. I tend to agree that the additional experiment TOBSY-INEPT in this situation will not yield significantly new insights
- the answer to point 13 and the revision of figure 4 is really helpful in improving readability and clarifying the reviewer's concern

Rebuttal letter and response of reviewers' comments

Reviewer #2 (Remarks to the Author):

Comment 1: Regarding the preparation and conditions used for the two methods, we appreciate the substantial effort invested in both methods and the structure determination. However, the primary argument of the paper is to compare and differentiate between reversible and irreversible fibrils, whereas the structure was obtained using two different methods. One cannot help but wonder if the observed differences arise from the preparation and conditions, which is often the case with amyloids.

We thank the reviewer for the comment. We believe that on this point there has been a basic miscommunication and that both the Reviewer and the Authors are actually trying to say precisely the same thing, which in essence can be summarized as such: «The very same protein (i.e. either Human Lysozyme or HEWL) can form either reversible or irreversible fibrils by folding differently, depending on the preparation conditions / thermal history path followed».

Of course, in order to form the two main different reversible/irreversible folds, the conditions must be changed. Specifically, for the case of commercial human lysozyme, we could fabricate the rigid fibrils at 100 mM DTT and flexible fibril at both 20 mM and 100 mM DTT but with different length of incubation times: by taking 100 mM as a benchmark condition, this proves that the incubation time, and not the buffer conditions, is enough to direct the re-folding of lysozyme onto either rigid/irreversible or flexible/reversible fibrils. The formation of rigid and flexible fibril can therefore be directed using exactly the same conditions (DTT, temperature, NaCl et al.) but different incubation times, as confirmed in the experiments. The same was observed for the case of commercial HEWL. Actually, small variations on the preparation protocol (85°C vs 80°C, compare Table S3) introduce no noticeable differences within the same class of amyloid and certainly do not affect the path towards reversible or irreversible fibrils; major differences in incubation time (3 hours vs 5 minutes) are needed to direct the re-folding into either class of amyloid, i.e. reversible or irreversible. Since we compare the atomic structure of the same protein sequence (human lysozyme) folded into either rigid/irreversible or flexible/reversible amyloid fibrils, the atomistic structural changes can be ascribed primarily to the different times of incubations followed at high temperature (3 hours vs 5 minutes) and only minimally to the differences between 85°C (for commercial Human lysozyme, rigid fibrils, cryoEM) vs 80°C (for purified Human lysozyme, flexible fibrils, ssNMR). Yet, the main point of the manuscript, as correctly noted by the Reviewer, is to compare the different atomic structures of the two different classes of amyloids from the same protein; if 5°C degrees difference in unfolding temperature also contribute together with the 36-fold difference in incubation time at high temperature, this is now duly acknowledged in the new Table S3 and can be considered as integrant part of the protocol followed to achieve the two different classes of amyloid folds. Yet, from the control experiments run at 85°C for the production of both flexible and rigid fibrils from commercial human lysozyme (see Table S3, cryoEM, AFM, CD), our understanding is that these 5 °C difference actually play no role at all.

The authors responded, stating, "Furthermore, we note that the preparation protocol and buffer for the preparation of the amyloid of human lysozyme used for the cryoEM (commercial) and ssNMR (expressed) studies were identical." However, upon comparing the buffer composition between the two versions, we noticed a discrepancy in the incubation temperature for the rigid fibrils of human lysozyme (see Table attached).

We thank the reviewer for bringing this to our attention. Actually, we were asked by another reviewer to provide more details on the protocol in the first round of revision, and we therefore completed the missing section of flexible fibrils preparation and made a correction on the temperatures used. Besides, we inspected the effects of the temperature on the fibril formation and found the rigid fibril of commercial human lysozyme can be formed in a range of temperature (80, 85 and 90°C). In the last revision, we reviewed again the many experiments we carried out and realized that the samples for cryoEM on human lysozyme were prepared at 85 °C and therefore made the correction from 80 to 85 °C in the last round of revision. By contrast, we observed that the commercial HEWL are more thermally stable and a slightly higher temperature range is needed to form rigid and flexible fibrils (85-90°C).

Nevertheless, we agree with the reviewer about the apparent discrepancy on the protocol. To eliminate this ambiguity, we have made a correction by providing the explicit values of the DTT and temperatures in the protocols. Inspired by the table kindly made by the reviewer, we also supplemented a table of the protocol of flexible and rigid fibrils prepared from human lysozyme and HEWL used in this work to make this protocol clearer to the readers. The corrections have been made in the method section and the Table S3 in the revised manuscript and SI and we are most thankful to the reviewer for having suggested these changes and the table.

Additionally, the literature (<https://doi.org/10.1016/j.bpc.2023.106962>) suggests that lysozyme fibrillation depends on DTT concentration. We observed that the DTT concentration changes between the rigid and flexible states, with no clear DTT concentration specified for the flexible fibrils.

We agree with the reviewer that DTT concentration is crucial for directing the fold towards reversible and irreversible fibrils. If the DTT concentration is not enough (4mM as in the literature suggested by the Reviewer) or DTT absent, full unfolding (a requirement for re-folding into irreversible amyloid fibrils) is not possible and irreversible fibrils cannot therefore be prepared. However, if DTT concentration is 20 mM or more, the quantity is sufficient to fully unfold the protein. In our experiments, we found no fibrils in absence of DTT and a lower quantity of fibril at low (2 mM) DTT concentration while we found that fibrils showed no difference in either quantity or morphology at both 20mM and 100mM DTT conditions. At these high DTT concentrations, reversible vs irreversible re-folding depends only on the time the proteins stay in the range 80-90°C. This is why no differences are observed whether 20 or 100 mM are used in the path followed for fibrillization. To be more explicit, we have made the modification in the text of the revised manuscript and in the Table S3 of the revised SI.

The authors also stated, "The atomic polymorphism of commercial and expressed human lysozyme is never compared directly." However, the subsequent conclusions are indeed compared. The authors assume that only the incubation time matters, suggesting in the final model that reversible morphologies form first, followed by irreversible fibrils at a later stage. Given the different buffers due to varying DTT concentrations, which can influence lysozyme's ability to form amyloid fibrils, time cannot be considered the sole parameter. Therefore, the assumption that reversible fibrils form first, followed by rigid ones, should be stated more cautiously, clarifying that the proposed mechanism is still under examination.

Following the point above, the fibrillization path becomes independent on DTT content provided we are above 20 mM. However, we agree that the statements must be made more cautiously: as indicated above, we have tried to be more explicit by correcting the method section and Table S3 in the revised manuscript and SI.

Comment 3: Thank you for clarifying the cryo-EM methods. It has become much clearer. However, there appears to be confusion between the generation of the first 3D volume (initial model) and model building, which involves generating the PDB file with the 3D alignment of the amino acid sequence.

The image processing section must include all data processing steps from motion correction to postprocessing, including the initial model, 3D classification, 3D refinement, CTF refinement, Bayesian polishing, and postprocessing. Therefore, the section from "Alignments of the segments from the selected 2D classes" to "such mask were used in a final postprocessing step" should be moved to the image processing section. Additionally, please correct the typo "mask were" to "mask was."

We thank the reviewer for suggesting this change. It has been made and we now refer to the EM "model" as a "volume" in order to remove the ambiguity with the structural coordinate-based model.

Comment 4: It is crucial to at least mention this impurity, which might affect the results concerning irreversible fibril formation. Even if the impurity was not detected in the previous study, it does not mean it was not present at very low concentrations. Furthermore, the minimum concentration required to trigger lysozyme irreversible fibril formation is unknown.

We shall note that we only studied the mesoscopic characterization by AFM in the previous work (rather than atomic level by cryoEM) and the secondary structure by FTIR and CD which are bulk methods; furthermore, no traces of impurities were detected by MALDI-TOF. As such, even if impurities were present (to undetectable levels) the conclusions drawn before (reference 24) remain valid (confirmed in the present study).

Concerning the presence of the impurity in the present study, we identified the contaminant as a rice protein (trypsin inhibitor) and its sequence fits perfectly the electron density map to the extent that the structure of the ensued amyloid is solved and will be the object of a follow up-manuscript. The Reviewer is concerned that this protein impurity may affect the results on irreversible fibril formation, and we can think of two ways this could happen: i) co-fibrillization of the two proteins into a single irreversible amyloid fibril or ii) seeding, nucleation (primary or secondary), or catalyzation of lysozyme irreversible fibrils by the presence of the trypsin inhibitor protein impurity. We can, however, rule out both scenarios based on the following observations:

a) For scenario i) since we solve structures of irreversible amyloids from both human lysozyme and trypsin inhibitor and match the primary protein sequences in both cases by such structure solving process, co-fibrillization events are never detected and thus can be dismissed;

b) For scenario ii) it is sufficient to note that reversible and irreversible fibrils formation following the protocol discussed in the manuscript, is observed also for both the human lysozyme in our previous article (reference 24 in the manuscript), where no impurities were detected by MALDI-TOF, and also in the case of HEWL. Thus, both types of fibrils (reversible and irreversible) can be formed via our protocol independently from the presence of a contaminant protein, for different batches (of the same protein) but also for different classes of lysozyme. In conclusion, we have no evidence of any sort that this impurity can affect the formation of irreversible lysozyme fibrils.

Nonetheless, we agree with the reviewer to mention the impurity of the commercial batch of human lysozyme from Sigma (see the method section in the revised manuscript), in order to avoid any possible ambiguity and also to anticipate the future paper on trypsin inhibitor amyloid fibrils.

BTW, if a comment is unclear, please contact the editor for clarification instead of making assumptions, as this ensures an accurate response. Regarding the polymorphism observed in the sample by cryo-EM, the key question is whether this polymorphism is due to incubation time, as suggested, or due to natural variability such as buffer composition or temperature. If the exact same conditions are used (which is not the case here), it becomes feasible to compare the reversible and irreversible states in the model suggested in Figure 5.

As indicated in the answers above, the polymorph structure is not the results from the incubation condition related to buffer and temperature, but essentially to the time spent at the temperature range 80-90°C, provided DTT is present at concentration of 20 mM or more.

Comment 18: The question was about the number of independent experiments. In the revised version, where you indicated the number of areas on the mica, you have a typo :” fibri”

Thank you for pointing out the typo. In the last revision, the data were collected from the different samples in the same experiment. We have now corrected it by collecting more than 700 fibrillar aggregates from each type of condition at least three independent experiments. The corresponding modifications have been made in Figure 1b and the method section.

Few more typos detected:

- Abstract: combing should be combining
- Fig S4 and S5: “the a map”.
- Methods: “For the expressed human lysozyme reversible fibrils used in solid-state NMR, the protein were purified.”

Thank you for pointing out these typos. They are now fixed in the revised manuscript and SI.

Reviewer #3 (Remarks to the Author):

The authors have addresses all of my comments and I think this revised manuscript is improved and ready for publication.

We are delighted to receive this positive comment and appreciate the reviewer input for having allowed improving the manuscript.

Reviewer #5 (Remarks to the Author):

The concerns of reviewer #1 in regard to NMR experiments have been thoroughly addressed. In terms of the technical description of the measurements and discussion, the manuscript is ready for publication.

In particular,

- point 7 has been clarified. I tend to agree that the additional experiment TOBSY-INEPT in this situation will not yield significantly new insights
- the answer to point 13 and the revision of figure 4 is really helpful in improving readability and clarifying the reviewer’s concern

We thank the reviewer for the positive assesemnt.

REVIEWERS' COMMENTS

Reviewer #2 (Remarks to the Author):

The authors have addressed the comments.